#### Recent increases in the atmospheric growth rate and emissions of HFC-23 1

#### (CHF<sub>3</sub>) and the link to HCFC-22 (CHClF<sub>2</sub>) production 2

Peter G. Simmonds<sup>1</sup>, Matthew Rigby<sup>1</sup>, Archie McCulloch<sup>1</sup>, Martin K. Vollmer<sup>2</sup>, Stephan 3

Henne<sup>2</sup>, Jens Mühle<sup>3</sup>, Simon O'Doherty<sup>1</sup>, Alistair J. Manning<sup>4</sup>, Paul B. Krummel<sup>5</sup>, Paul J. 4

Fraser<sup>5</sup>, Dickon Young<sup>1</sup>, Ray F. Weiss<sup>3</sup>, Peter K. Salameh<sup>3</sup>, Christina M. Harth<sup>3</sup>, Stefan 5

- Reimann<sup>2</sup>, Cathy M. Trudinger<sup>5</sup>, L. Paul Steele<sup>5</sup>, Ray H. J. Wang<sup>6</sup>, Diane J. Ivy<sup>7</sup>, Ronald G. 6
- Prinn<sup>7</sup>, Blagoj Mitrevski<sup>5</sup>, and David M. Etheridge<sup>5</sup>. 7
- 8
- <sup>1</sup>Atmospheric Chemistry Research Group, University of Bristol, Bristol, UK 9
- <sup>2</sup> Swiss Federal Laboratories for Materials Science and Technology, Laboratory for Air Pollution 10 and Environmental Technology (Empa), Dübendorf, Switzerland,
- 11
- 12 <sup>3</sup> Scripps Institution of Oceanography (SIO), University of California at San Diego, La Jolla, California, USA 13
- <sup>4</sup> Met Office Hadley Centre, Exeter, UK 14
- <sup>5</sup>Climate Science Centre, Commonwealth Scientific and Industrial Research Organisation 15
- (CSIRO) Oceans and Atmosphere, Aspendale, Victoria, Australia 16
- 17 <sup>6</sup> School of Earth, and Atmospheric Sciences, Georgia Institute of Technology, Atlanta, Georgia, 18 USA
- <sup>7</sup> Center for Global Change Science, Massachusetts Institute of Technology, Cambridge, 19
- Massachusetts, USA 20
- 21

22 Correspondence to: P.G. Simmonds (petergsimmonds@aol.com)

- 23
- 24 Abstract

High frequency measurements of trifluoromethane (HFC-23, CHF<sub>3</sub>), a potent hydrofluorocarbon 25

greenhouse gas, largely emitted to the atmosphere as by-product of production of the 26

hydrochlorofluorocarbon HCFC-22 (CHClF<sub>2</sub>), at five core stations of the Advanced Global 27

Atmospheric Gases Experiment (AGAGE) network, combined with measurements on firn air, 28

old Northern Hemisphere air samples and Cape Grim Air Archive (CGAA) air samples, are used 29

- to explore the current and historic changes in the atmospheric abundance of HFC-23. These 30
- measurements are used in combination with the AGAGE 2-D atmospheric 12-box model and a 31
- Bayesian inversion methodology to determine model atmospheric mole fractions and the history
- of global HFC-23 emissions. The global modelled annual mole fraction of HFC-23 in the 33

background atmosphere was  $28.9 \pm 0.6$  pmol mol<sup>-1</sup> at the end of 2016, representing a 28% 34

increase from  $22.6 \pm 0.4$  pmol mol<sup>-1</sup> in 2009. Over the same time frame, the modelled mole 35

- fraction of HCFC-22 increased by 19% from  $199 \pm 2$  pmol mol<sup>-1</sup> to  $237 \pm 2$  pmol mol<sup>-1</sup>. 36
- However, unlike HFC-23, the annual average HCFC-22 growth rate slowed from 2009 to 2016 37
- at an annual average rate of -0.5 pmol mol<sup>-1</sup> yr<sup>-2</sup>. This slowing atmospheric growth is consistent 38
- with HCFC-22 moving from dispersive (high fractional emissions) to feedstock (low fractional 39
- emissions) uses, with HFC-23 emissions remaining as a consequence of incomplete mitigation
- from all HCFC-22 production. 41

Our results demonstrate that, following a minimum in HFC-23 global emissions in 2009 of 9.6  $\pm$ 0.6 Gg yr<sup>-1</sup>, emissions increased to a maximum in 2014 of  $14.5 \pm 0.6$  Gg yr<sup>-1</sup> and then declined 44 to  $12.7 \pm 0.6$  Gg yr<sup>-1</sup> (157 Mt CO<sub>2</sub>-e yr<sup>-1</sup>) in 2016. The 2009 emissions minimum is consistent 45 with estimates based on national reports and is likely a response to the implementation of the 46 Clean Development Mechanism (CDM) to mitigate HFC-23 emissions by incineration in 47 developing (Non-Annex 1) countries under the Kyoto Protocol. Our derived cumulative 48 emissions of HFC-23 during 2010-2016 were  $89 \pm 2$  Gg (1.1  $\pm$  0.2 Gt CO<sub>2</sub>-e), which led to an 49 increase in radiative forcing of  $1.0 \pm 0.1$  mW m<sup>-2</sup> over the same period. Although the CDM had 50 reduced global HFC-23 emissions, it cannot now offset the higher emissions from increasing 51 HCFC-22 production in Non-Annex 1 countries, as the CDM was closed to new entrants in 52 53 2009. We also find that the cumulative European HFC-23 emissions from 2010 to 2016 were ~1.3 Gg, corresponding to just 1.5% of cumulative global HFC-23 emissions over this same 54 period. The majority of the increase in global HFC-23 emissions since 2010 is attributed to a 55 delay in the adoption of mitigation technologies, predominantly in China and east Asia. 56 However, a reduction in emissions is anticipated, when the Kigali 2016 amendment to the 57 58 Montreal Protocol, requiring HCFC and HFC production facilities to introduce destruction of HFC-23, is fully implemented. 59

### 61 **1.** Introduction

Due to concerns about climate change, trifluoromethane (CHF<sub>3</sub>, HFC-23) has attracted 62 interest as a potent greenhouse gas due to a100-yr integrated global warming potential (GWP) of 63 64 12400 (Myhre et al., 2013) and an atmospheric lifetime of ~228 yr (SPARC, 2013). Hydrofluorocarbons (HFCs) were introduced as replacements for ozone-depleting 65 chlorofluorocarbons (CFCs) and hydrochlorofluorocarbons (HCFCs) - for example, HFC-134a 66 as a direct replacement for CFC-12 (Xiang et al., 2014). However, HFC-23 is a by-product of 67 chlorodifluoromethane HCFC-22 (CHClF<sub>2</sub>) production, resulting from the over-fluorination of 68 chloroform (CHCl<sub>3</sub>). Most HFC-23 has historically been vented to the atmosphere (UNEP, 69 2017a). HFC-23 has also been used as a feed-stock in the production of Halon-1301 (CBrF<sub>3</sub>) 70 (Miller and Batchelor, 2012) which substantially decreased with the phase-out of halons in 2010 71 under the Montreal Protocol, a landmark international agreement designed to protect the 72 stratospheric ozone layer. HFC-23 also has minor emissive uses in air conditioning, fire 73 extinguishers, and semi-conductor manufacture (McCulloch and Lindley, 2007) and very minor 74 emissions from aluminium production (Fraser et al., 2013). For developed countries HFC-23 75 emissions were controlled as part of the "F-basket" under the Kyoto Protocol, an international 76

treaty among industrialized nations that sets mandatory limits on greenhouse gas emissions,

(<u>https://ec.europa.eu/clima/policies/f-gas\_en</u>).

In the context of this paper, we discuss "developed" and "developing" countries which
we take to be synonymous with Annex 1 (Non-Article 5) countries and Non-Annex 1 (Article 5)
countries, respectively.

There have been a number of previous publications related to HFC-23. Oram et al. 82 (1998) measured HFC-23 by gas chromatography-mass spectrometry (GC-MS) analysis of Cape 83 84 Grim flask air samples and sub-samples of the Cape Grim Air Archive (CGAA) from 1978-1995 and reported a dry-air southern hemispheric atmospheric abundance of 11 pmol mol<sup>-1</sup> in late-1995. 85 Culbertson et al. (2004) estimated global emissions of HFC-23 using a one-box model and GC-MS 86 87 analysis of north American and Antarctic air samples. A top-down HFC-23 emissions history and a comprehensive bottom-up estimate of global HFC-23 emissions were reported by Miller et al. 88 (2010) using Advanced Global Atmospheric Gases Experiment (AGAGE) observations (2007-89 90 2009) and samples from the CGAA (1978-2009). Montzka et al. (2010), using measurements of firn air from the permeable upper layer of an ice sheet and ambient air collected during three 91 expeditions to Antarctica between 2001 and 2009, constructed a consistent Southern Hemisphere 92 (SH) atmospheric history of HFC-23 that was reasonably consistent with Oram et al., (1998) 93 results. Kim et al., (2010), reported HFC-23 measurements (November 2007-December 2008) 94 from Gosan, Jeju Island, South Korea and also estimated regional atmospheric emissions. Asian 95 emissions of HFC-23, including those for China have been reported by Yokouchi et al., 2006; 96 Stohl et al., 2010; Li et al., 2011; Yao et al., 2012. Most recently, Fang et al., (2014, 2015) have 97 provided bottom-up and top-down estimates of HFC-23 emissions from China and east Asia and 98 included observed HFC-23 mixing ratios at three stations - Gosan, South Korea, and Hateruma 99 and Cape Ochi-ishi, Japan. Remote sensing observations of HFC-23 in the upper troposphere 100 101 and lower stratosphere by two solar occultation instruments have also been reported (Harrison et al., 2012), indicating an abundance growth rate of  $5.8 \pm 0.3\%$  per year, similar to the CGAA 102 103 surface trend of  $5.7 \pm 0.4\%$  per year over the same period (1989-2007).

HCFC-22 is used principally in air-conditioning and refrigeration, with minor uses in
foam blowing and as a chemical feedstock in the manufacture of fluoropolymers, such as
polytetrafluorethylene (PTFE). HCFC-22 production and consumption (excluding feedstock use)
are controlled under the Montreal Protocol. We have previously reported on the changing trends
and emissions of HCFC-22 (Simmonds et al., 2017 and references therein).

Technical solutions to mitigate HFC-23 emissions have included optimisation of the
 HCFC-22 production process and voluntary and regulatory capture and incineration in
 developed (Annex 1) countries (McCulloch, 2004). Mitigation in developing countries (Non-

Annex 1) has been introduced under the United Nations Framework Convention on Climate 112 Change (UNFCCC) Clean Development Mechanism (CDM) to destroy HFC-23 from HCFC-22 113 production facilities (UNEP, 2017a). This allowed certain HCFC-22 production plants in 114 developing countries to be eligible to provide Certified Emission Reduction (CER) credits for 115 the destruction of the co-produced HFC-23. Beginning in 2003, there were 19 registered HFC-23 116 incineration projects in five developing countries with the number of projects in each country 117 shown in parenthesis: China (11), India (5), Korea (1), Mexico (1) and Argentina (1) (Miller et 118 al., 2010). The first CER credits under the CDM for HFC-23 abatement in HCFC-22 plants were 119 approved in 2003 with funding through 2009. However, the CDM projects covered only about 120 half of the HCFC-22 production in developing countries. The substantial reduction in global 121 HFC-23 emissions during 2007-2009 was attributed by Miller et al. (2010) to the destruction of 122 HFC-23 by CDM projects. In a subsequent paper, Miller and Kuijpers (2011) predicted future 123 increases in HFC-23 emissions by considering three scenarios: a reference case with no 124 additional abatement, and two opposing abatement measures, less mitigation and best practice 125 involving increasing application of mitigation through HFC-23 incineration. Historically there 126 has been a lack of information about HFC-23 emissions from the non-CDM HCFC-22 127 production plants, although Fang et al. (2015) provided a top-down estimate of HFC-23/HCFC-128 22 co-production ratios in non-CDM production plants. They reported that the HFC-23/HCFC-129 22 co-production ratios in all HCFC-22 production plants were  $2.7\% \pm 0.4$  by mass in 2007, 130 131 consistent with values reported to the Executive Committee of the Montreal Protocol (UNEP, 132 2017a).

Here, we use the high frequency atmospheric observations of HFC-23 and HCFC-22 133 abundances measured by (GC-MS) at the five longest-running remote sites of the Advanced 134 Global Atmospheric Gases Experiment (AGAGE). The site coordinates and measurement time 135 frames of HFC-23 and HCFC-22 are listed in Table 1. To extend our understanding of the long-136 term growth rate of HFC-23, we combine the direct AGAGE atmospheric observations with 137 results from an analysis of firn air collected in Antarctica and Greenland, a series of old 138 Northern Hemisphere air samples, and archived air from the CGAA (Fraser et al. (1991). The 139 AGAGE 2-D 12-box model and a Bayesian inversion technique are used to produce global 140 141 emission estimates for HFC-23 and HCFC-22 (Cunnold et al., 1983; Rigby et al., 2011, 2014). We also include observations from the AGAGE Jungfraujoch station to determine estimates of 142 143 European HFC-23 emissions (see section 3.3).

### 147 **2.** Methodology

### 148 2.1. AGAGE Instrumentation and Measurement Techniques

Ambient air measurements of HFC-23 and HCFC-22 at each site are made using the AGAGE GC-MS-Medusa instrument which employs an adsorbent-filled (HayeSep D) microtrap cooled to ~ -175°C to pre-concentrate the analytes during sample collection from 2 litres of air (Miller et al., 2008; Arnold et al., 2012). Samples are analysed approximately every 2 hours and are bracketed by measurements of quaternary standards to correct for short-term drifts in instrument response. Additional details of the analytical methodology are provided in the Supplementary material (1).

2.2. Firn and Archived Air

We used air samples from firn and archived in canisters to reconstruct an atmospheric 157 HFC-23 history. Firn air samples from Antarctica and Greenland were analysed for HFC-23 158 using the same technology as the *in situ* measurements; details are provided in the 159 Supplementary Material (2). The Antarctic samples were collected at the DSSW20K Law Dome 160 site in 1997-1998 (Trudinger et al., 2002) and include one deep sample from the South Pole 161 collected in 2001 (Butler et al., 2001). Greenland samples were collected at the NEEM (North 162 Greenland Eemian Ice Drilling) site in 2008 (Buizert et al., 2012). The CSIRO firn model 163 (Trudinger et al., 1997, 2013) was used to derive age spectra for the individual firn samples and 164 more details on these samples and their analysis are given in Vollmer et al. (2016, 2018) and 165 Trudinger et al. (2016). A diffusion coefficient of HFC-23 relative to CO<sub>2</sub> of 0.797 was used 166 (Fuller et al., 1966). 167

CGAA measurements from three separate analysis periods were also used for the 168 reconstruction of past HFC-23 abundances. The CGAA samples have been collected since 1978 169 at the Cape Grim Air Pollution Station and amount to >130 samples, the majority in internally 170 electropolished stainless steel canisters (Fraser et al., 2016, 2017). Samples were analysed under 171 varying conditions in 2006, 2011 and 2016. Here, we use a composite of the results from these 172 measurement sets. Details are given in the Supplementary Material (2) and by Vollmer et al. 173 174 (2018). A series of old Northern Hemisphere (NH) air samples were also measured together with the measurements of the CGAA samples at the Scripps Institution of Oceanography see 175 176 Supplementary Material (2) and Mühle et al., (2010; Vollmer et al., (2016).

2.3. Calibration Scales

The estimated accuracies of the calibration scales for HFC-23 and HCFC-22 are

discussed below and a more detailed discussion of the measurement techniques and calibration

procedures are reported elsewhere (Miller et al., 2008; O'Doherty et al., 2009; Mühle et al., 2010). HFC-23 and HCFC-22 measurements from all AGAGE sites are reported relative to the SIO-07 and SIO-05 primary calibration scales respectively, which are defined by suites of standard gases prepared by diluting gravimetrically prepared analyte mixtures in  $N_2O$  to nearambient levels in synthetic air (Prinn et al., 2000; Miller et al., 2008).

The absolute accuracies of these primary standard scales are uncertain because possible 185 systematic effects are difficult to quantify or even identify. Combining known statistical and 186 estimated systematic uncertainties, such as measurement and propagation errors, and quoted 187 reagent purities, generally yields lower uncertainties than are supported by comparisons among 188 independent calibration scales (Hall et al., 2014). Furthermore, some systematic uncertainties 189 may be normally distributed, while others, like reagent purity, are skewed in one direction. 190 Estimates of calibration accuracies and their uncertainties are nevertheless needed for 191 192 interpretive modelling applications. So, despite the difficulty in estimating unknown uncertainties, it is incumbent on those responsible for the measurements to provide an overall 193 194 assessment of accuracy. Accordingly, we liberally estimate the absolute accuracies of these 195 measurements as -3% to +2% for HFC-23 and  $\pm1\%$  for HCFC-22. The larger and asymmetric uncertainty for HFC-23 is due to its lower atmospheric and standard concentration, and to the 196 lower stated purity of the HFC-23 reagent used to prepare the primary calibration scale. 197

### 199 2.4. Selection of baseline data (unpolluted background air)

Baseline in situ monthly mean HFC-23 and HCFC-22 mole fractions were calculated by 200 excluding values enhanced by local and regional pollution influences, as identified by the 201 iterative AGAGE pollution identification algorithm (for details, see Appendix in O'Doherty et 202 al., 2001). Briefly, baseline measurements are assumed to have Gaussian distributions around the 203 local baseline value, and an iterative process is used to filter out the data that do not conform to 204 this distribution. A second-order polynomial is fitted to the subset of daily minima in any 121-205 day period to provide a first estimate of the baseline and seasonal cycle. After subtracting this 206 polynomial from all the observations, a standard deviation and median are calculated for the 207 residual values over the 121-day period. Values exceeding three standard deviations above the 208 baseline are assigned as non-baseline (polluted) and removed from further consideration. The 209 process is repeated iteratively to identify and remove additional non-baseline values until the 210 new and previous calculated median values agree within 0.1%. 211

### 213 2.5. Bottom-up emissions estimates

The sources of information on production and emissions of HFCs are generally incomplete 214 and do not provide a comprehensive database of global emissions. In Supplementary Material 215 (3), we compile global HCFC-22 production and HFC-23 emissions data. HCFC-22 is used in 216 two ways: (1) dispersive applications, such as refrigeration and air conditioning, whose 217 production is controlled under the Montreal Protocol and reported by countries as part of their 218 total HCFC production statistics, and (2) feedstock applications in which HCFC-22 is a reactant 219 in chemical processes to produce other products. Although there is an obligation on countries to 220 221 report HCFC-22 feedstock use to UNEP, this information is not made public. HCFC-22 production for dispersive uses was calculated from the UNEP HCFC database (UNEP, 2017b) 222 and the Montreal Protocol Technology and Economic Assessment Panel 2006 Assessment 223 (TEAP, 2006). Production for feedstock use was estimated using trade literature as described in 224 the Supplementary Material (3) and the sum of production for dispersive and feedstock uses is 225 shown in Table S3. 226

HFC-23 emissions from Annex 1 countries are reported as a requirement of the 227 UNFCCC. Table S4 shows the total annual HFC-23 emissions reported by these countries 228 (UNFCCC, 2017). There is a small uncertainty in these UNFCCC emissions due to whether 229 countries report on a calendar or fiscal year basis. The data include emissions from use of HFC-230 23 in applications such as semi-conductor manufacture and fire suppression systems. These 231 232 minor uses of HFC-23, originally produced in a HCFC-22 plant, will result in the eventual emission of most or all into the atmosphere and emissions have remained relatively constant at 233  $0.13 \pm 0.01$  Gg yr<sup>-1</sup>, a maximum of 10% of all emissions (UNFCCC, 2017). Non-Annex 1 234 countries listed in Table S4 were eligible for financial support for HFC-23 destruction under the 235 CDM. Their emissions were calculated by applying factors to their estimated production of 236 HCFC-22 and offsetting this by the amount destroyed under CDM, as described in the 237 Supplementary Material (3). We discuss these independent emission estimates because they are 238 useful as *a priori* data constraints ("bottom-up" emission estimates) which we compare to 239 observation-based "top-down" estimates. 240

2.6. Global atmospheric model

Emissions were estimated using a Bayesian approach in which our *a priori* estimates of the emissions growth rate were adjusted by comparing modelled baseline mole fractions to the atmospheric observations (Rigby et al., 2011, 2014). The firn air measurements were included in the inversion, with the age spectra from the firn model used to relate the firn measurements to high-latitude atmospheric mole fractions (Trudinger et al., 2016; Vollmer et al., 2016). A 12-box 247 model of atmospheric transport and chemistry was used to simulate baseline mole fractions. which assumed that the atmosphere was divided into four zonal bands (90-30°N, 30-0°N, 0-30°S, 248 30-90°S) and at 500hPa and 200hPa vertically (Cunnold et al., 1994; Rigby et al., 2013). The 249 model uses an annually repeating, monthly varying hydroxyl radical (OH) field from 250 Spivakovsky et al (2000), which has been adjusted to match the observed trend in methyl 251 chloroform (e.g. Rigby et al. 2013). For the gases in this paper, potential variations in OH 252 concentration (e.g. Rigby et al., 2017) were not found to lead to a large change in the derived 253 emissions (see Supplementary Material 4). Annually repeating, monthly varying transport 254 parameters were used as described in Rigby et al., (2014). The temperature-dependent rate 255 constant for reaction with OH was taken from Burkholder et al. (2015), which led to a lifetime of 237 256 yr for HFC-23 and 11.6 yr for HCFC-22. As in previous publications (e.g. Rigby et al., 2014), 257 258 uncertainties in the monthly mean baseline observations in each semi-hemisphere were taken to be the quadratic sum of the measurement repeatability and the variability of the observations within the month 259 that were flagged as "baseline", using the method in O'Doherty et al. (2001). The variability was used 260 261 to approximate model uncertainty, as it was assumed to be a measure of the time scales not 262 resolved by the model. No correlated uncertainties were assumed in the model-measurement mismatch 263 uncertainty and both the model-measurement mismatch and the *a priori* constraint were assumed to be 264 described by Gaussian probability density functions. Seasonal emissions estimates in each semi-265 hemisphere were derived in the inversion. The inversion propagates uncertainty estimates from the 266 measurements, model and prior emissions growth rate to these *a posteriori* emissions estimates. The prior 267 emissions growth rate uncertainty was somewhat arbitrarily chosen at a level of 20% of the maximum a priori emissions and no correlation was assumed between prior estimates of annual emissions growth 268 269 rate. In contrast to Rigby et al. (2014), in which a scaling factor of the emissions was solved for in the 270 inversion, here we determined absolute emissions, which were found to lead to more robust uncertainty 271 estimates when emissions were very low. A posteriori emissions uncertainties were augmented with an estimate of the influence of uncertainties in the lifetime, as described in Rigby et al. (2014). 272

273

### 274 2.7. Regional scale atmospheric inversion

HFC-23 pollution events are still observed at sites in north east Asia and Europe. The
former have recently been used in regional scale inverse modelling studies to derive emission
estimates for the Asian region and these results are summarised in Section 3.3. In contrast, little
attention has recently been given to HFC-23 emissions from Europe. We examine European
emissions, using a regional scale inversion tool based on source sensitivities as estimated by a
Lagrangian Particle Dispersion Model run in backward mode, combined with a Bayesian
inversion framework.

Surface source sensitivities were computed with the Lagrangian Particle Dispersion Model (LPDM) FLEXPART (Stohl et al., 2005) driven by operational analysis/forecasts from the European Centre for Medium-Range Weather Forecasts (ECMWF) Integrated Forecasting System (IFS) modelling system with a horizontal resolution of 0.2° x 0.2° for central Europe and 1° x 1° elsewhere. 50,000 model particles were released for each 3-hourly time interval and followed backward in time for 10 days.

### 291 2.7.2. Bayesian inversion framework

A spatially resolved, regional-scale emission inversion, using the FLEXPART-derived 292 source sensitivities and a Bayesian approach was applied to estimate European HFC-23 annual 293 emissions for individual years between 2009 and 2016. The details of the inversion method were 294 recently published in estimating Swiss methane emissions (Henne et al., 2016). This inversion 295 methodology was part of an HFC inversion inter-comparison (Brunner et al, 2017) and was 296 applied to HFC and HCFC emissions in the eastern Mediterranean (Schönenberger et al., 2017). 297 Here, the inversion relies on the continuous observations from the Jungfraujoch and Mace Head 298 and requires *a priori* estimates of the emissions distribution. The observations are split into a 299 baseline concentration and above-baseline excursions of the signal that are attributed to recent 300 301 emissions using the method of Ruckstuhl et al. (2012). The inversion estimates spatially distributed, annual mean emissions and a two-weekly concentration baseline. In the case of 302 HFC-23 the baseline concentration is very well defined due to the relatively infrequent 303 occurrence of larger pollution events. The inversion results were not significantly different when 304 305 the baseline was not updated as part of the inversion. The spatial distribution was solved on a grid with different sized rectangular cells. The grid size was inversely proportional to annual 306 307 total source sensitivities and, therefore, was finer close to the measurement sites and coarser in more remote regions that seldom influence the sites. In contrast to previous applications, the grid 308 309 resolution was also increased around likely point emitters in order to better localise these potentially large contributors. 310

In this study the inversion was set up using complete covariance matrices. We designed the *a priori* covariance matrix in such a way that the total *a priori* uncertainty for each of the regions/countries was 200% and proportional to the emissions in each inversion grid cell. Offdiagonal elements of the matrix were filled with the assumption of exponentially decaying spatial correlation of the uncertainties with a length scale of 10 km. The choice of this rather 316 small spatial correlation scale was motivated by the assumed strong contribution from point source emissions, which should result in spatially rather uncorrelated *a priori* uncertainties. 317 The data-mismatch covariance matrix contained uncertainty elements that describe the 318 uncertainty of the observations and the transport model. The observation uncertainty was taken 319 from target gas measurements, whereas the model uncertainty was estimated as the RMSE (root 320 mean square) of the *a priori* simulations Henne et al., (2016). The off-diagonal elements of the 321 covariance matrix were again assumed to exponentially decay with time between the data points. 322 The resulting correlation time scale was estimated separately for each site from a fit to the auto-323 correlation function of the prior model residuals (see Schönenberger et al., 2017) and was in the 324 order of 0.2 to 0.3 days. 325

### 327 2.7.3. *A priori* emissions and sensitivity inversions

Spatially distributed *a priori* emissions were generated from individual national inventory 328 reports (NIR) to UNFCCC (2017). Most European countries separately list the emissions of 329 HFC-23 by sector in Table 2(II) of their submissions. Here, we chose two different approaches 330 to spatially distribute these bottom-up estimates and use these as input for two sensitivity 331 332 inversions. In the first approach (UNFCCC\_org), we directly follow the categorisation in each NIR and assign emissions from 'Fluorochemical production' to individual production sites as 333 taken from Keller et al. (2011), and shown in Figure S4, whereas emissions from 'Electronics 334 industry' and 'Product use' were distributed according to population density (Center for 335 International Earth Science Information Network - CIESIN - Columbia University. 2016, 336 available at https://ciesin.columbia.edu/data/hrsl/). Countries reporting no or zero HFC-23 337 emissions were assigned a per capita emission factor equal to 1/10 of the average per capita 338 emission factor from reporting countries. This mostly impacts countries at the periphery of the 339 inversion domain. In our second approach (UNFCCC\_r0.5), we used the same spatial 340 disaggregation as before, but assigning 50% of the 'Electronics industry' and 'Product use' 341 emissions in each country to the likely point source locations and distribute the remainder by 342 population. Inversions using the HFC-23 inventory provided by EDGAR (version 4.2) as a 343 priori were tested. However, these inversions showed much weaker model performance than 344 345 those based on UNFCCC priors and, hence, were dropped from any further analysis.

3.

**Results and Discussion** 

### 350 3.1. Atmospheric mole fractions

Figure 1 shows the HFC-23 modelled mole fraction for the four equal-mass latitudinal 351 subdivisions of the global atmosphere calculated from the 12-box model and the combined GC-352 MS-Medusa in situ measurements (2008-2016), firn air data, old NH air and CGAA data. The 353 lower box shows the annual growth rates in pmol mol<sup>-1</sup> yr<sup>-1</sup>. We find that the global modelled 354 annual mole fraction of HFC-23 in the background atmosphere reached  $28.9 \pm 0.6$  pmol mol<sup>-1</sup>, 355 (1 $\sigma$  confidence interval) in December 2016, a 163% increase from 1995 and a 28% increase 356 from the 22.6  $\pm 0.2$  pmol mol<sup>-1</sup> reported in 2009 (Miller et al., 2010). In 2008 the annual mean 357 mid-year growth rate of HFC-23 was 0.78 pmol mol<sup>-1</sup> yr<sup>-1</sup>. By mid-2009 the growth rate 358 decreased to 0.68 pmol mol<sup>-1</sup>yr<sup>-1</sup>, rising to a maximum of 1.05 pmol mol<sup>-1</sup> yr<sup>-1</sup> in early 2014, 359 followed by a smaller decrease to 0.95 pmol mol<sup>-1</sup> yr<sup>-1</sup> in 2016. The growth rate of HFC-23 360 increased by 22 % from 2008 to 2016. In the Figure 1 inset, we compare the annual mean mole 361 fractions of HFC-23 and HCFC-22 recorded at Mace Head and Cape Grim, as examples of mid-362 latitude northern hemisphere (NH) and southern hemisphere (SH) sites, illustrating the site 363 divergence for these two compounds beginning around 2010. 364

Figure 2 shows our HCFC-22 modelled mole fractions for the four equal-mass latitudinal 365 subdivisions of the global atmosphere calculated from the 12-box model and the lower box 366 shows the HCFC-22 annual growth rates in pmol mol<sup>-1</sup> yr<sup>-1</sup>. The global modelled HCFC-22 367 annual mixing ratio in the background atmosphere reached peaked at  $238 \pm 2$  pmol mol<sup>-1</sup> in 368 December 2016, following the decline in the annual average global growth rate of HCFC-22 369 from 2008 to 2016 of 0.5 pmol mol<sup>-1</sup> yr<sup>-2</sup>. This decline in the global growth rate of HCFC-22 370 coincides with the phase out of HCFC production/consumption mandated by the 2007 371 amendment to the Montreal Protocol for Annex 1 countries, covering dispersive applications, but 372 not the non-dispersive use of HCFC-22 as a feedstock in fluoropolymer manufacture (UNEP, 373 2017a). Nevertheless, HCFC-22 remains the dominant HCFC in the atmosphere and accounts for 374 79% by mass of the total global HCFC emissions (Simmonds et al., 2017). In contrast to the 375 increasing growth rate of HFC-23, the growth rate of HCFC-22 has exhibited a steep 53% 376 decline from a maximum in January 2008 of 8.2 pmol mol<sup>-1</sup>yr<sup>-1</sup> to 3.8 pmol mol<sup>-1</sup> yr<sup>-1</sup> in 377 December 2016, further illustrating the divergence between these two gases between 2008 and 378 2016 (compare lower boxes in Figures 1 and 2). These results are an update of our previously 379 reported analysis AGAGE HCFC-22 data (Simmonds et al., 2017) for the period 1995-2015. 380

Miller et al. (2010) calculated global emissions of HFC-23 using the same AGAGE 12-384 box model as used here, but with a different Bayesian inverse modelling framework. Following a 385 peak in emissions in 2006 of 15.9 (+1.3/-1.2) Gg yr<sup>-1</sup>, modelled emission estimates of HFC-23 386 declined rapidly to 8.6 (+0.9/-1.0) Gg yr<sup>-1</sup> in 2009, which Miller noted was the lowest annual 387 emission for the previous 15 years. Based on the analysis of firn air samples and ambient air 388 measurements from Antarctica, Montzka et al. (2010) reported global HFC-23 emissions of 13.5 389  $\pm 2$  Gg yr<sup>-1</sup> (200  $\pm 30$  Mt CO<sub>2</sub>-e yr<sup>-1</sup>), averaged over 2006-2008. In Carpenter and Reimann 390 (2014), global emissions of HFC-23 estimated from measured and derived atmospheric trends 391 reached a maximum of 15 Gg yr<sup>-1</sup> in 2006, declined to 8.6 Gg in 2009, and subsequently 392 increased again to 12.8 Gg in 2012 (Figure 1-25, update of Miller et al., 2010 and Montzka et al., 393 2010). 394

The model derived HFC-23 emissions from this study, shown in Figure 3 and listed in 395 Table 2, reached an initial maximum in 2006 of  $13.3 \pm 0.8$  Gg yr<sup>-1</sup>, then declined steeply to  $9.6 \pm$ 396 0.6 Gg yr<sup>-1</sup> in 2009. Our HFC-23 emissions estimates, which include firn data and NH archive 397 air samples and a slightly different inverse method, are slightly lower in 2006 and slightly higher 398 in 2009 than the HFC-23 estimates of Miller et al. (2010) and Carpenter and Reimann (2014) 399 respectively. Our mean annual (2006-2008) HFC-23 emissions of 12.1 ( $\pm$  0.7) Gg yr<sup>-1</sup> are lower 400 than the Montzka et al., (2010) emissions estimates of  $13.5 \pm 2$  Gg yr<sup>-1</sup>, but agree within 401 uncertainties. However, our HFC-23 emissions then grew rapidly reaching a new maximum of 402  $14.5 \pm 0.6 \text{ Gg yr}^{-1}$  (180 ± 7 Mt CO<sub>2</sub>-eq yr<sup>-1</sup>) in 2014, only to decline again to  $12.7 \pm 0.6 \text{ Gg yr}^{-1}$ 403  $(157 \pm 7 \text{ Mt CO}_2\text{-eq yr}^{-1})$  in 2016. Cumulative HFC-23 emissions estimates from 2010 to 2016 404 were  $89 \pm 16$  Gg ( $1.1 \pm 0.2$  Gt CO<sub>2</sub>-e), contributing to an increase in radiative forcing of  $1.0 \pm$ 405  $0.1 \text{ mW m}^{-2}$ . 406

The global emission estimates for HCFC-22 are plotted in Figure 4 together with the corresponding Miller et al. (2010) emissions up to 2008 and the WMO 2014 emissions estimates (Carpenter and Reimann, 2014). Table 3 lists the global emission estimates, mole fractions and growth rate of HCFC-22. Our modelled HCFC-22 emissions reached a maximum global maximum of  $385 \pm 41$  Gg yr<sup>-1</sup> (696  $\pm$  74 Mt CO<sub>2</sub>-eq yr<sup>-1</sup>) in 2010, followed by a slight decline to  $370 \pm 46$  Gg yr<sup>-1</sup> (670  $\pm$  83 Mt CO<sub>2</sub>-eq yr<sup>-1</sup>) in 2016 at an annual average rate of 2.3 Gg yr<sup>-1</sup>.

413 3.3. Regional emissions

Several papers report Asian emissions of HFC-23, including those for China (Yokouchi et
al., 2006; Stohl et al., 2010; Kim et al., 2010; Li et al., 2011; Yao et al., 2012). Recent papers by
Fang et al. (2014, 2015) noted inconsistencies between the various bottom-up and top-down

- emissions estimates and provided an improved bottom-up inventory and a multi-annual top-
- down estimate of HFC-23 emissions for east Asia. They showed that China contributed 94-98 %
- of all HFC-23 emissions in east Asia and was the dominant contributor to global emissions:  $20 \pm$
- 6 % in 2000 rising to  $77 \pm 23$  % in 2005. China's annual HFC-23 top-down emissions in 2012
- were estimated at  $8.8 \pm 0.8$  Gg yr<sup>-1</sup> (Fang et al., 2015), 69% of our 2012 global emissions
- estimate of  $12.9 \pm 0.6$  Gg yr<sup>-1</sup>, listed in Table 2.

Based on the bottom-up estimated global HFC-23 emissions (shown in Supplementary 423 Material (3), Table S4), we show in Figure 5a global and Chinese emissions and the percentage 424 of Chinese emissions contributing to the global total. These bottom-up estimates show that a 425 steadily increasing fraction of global total HFC-23 emissions can be attributed to China, 426 averaging about 88% in 2011-2014. This rise is consistent with the increase in our calculated 427 bottom-up HCFC-22 production data compiled from industry sources and listed in 428 Supplementary Material (3), Table S3. Figure 5b, shows the bottom-up estimates of global and 429 China HCFC-22 production and the percentage contribution of China production to the global 430 total, further illustrating the dominance of HCFC-22 production in China. 431

Clearly China is the major contributor to recent global HFC-23 emissions which implies only 432 minor contributions from other regional emitters. However, HFC-23 pollution events are still 433 observed at our European sites. Keller et al. (2011) reported European HFC-23 emissions based 434 435 on inverse modelling for the period summer 2008 to summer 2010, assigning most of the emissions to point sources at HCFC-22 production sites. Here, we re-evaluated European HFC-436 23 emissions for the period 2009 to 2016 (see section 2.7). The key results from this analysis are 437 summarised in Figures 6 and in Table 4 and more detailed results are available in the 438 439 Supplementary Material (5).

Based on these inversion results, European emissions of HFC-23, though small on a 440 global scale were, in general, larger than reported to UNFCCC and exhibited considerable year-441 to-year variability (Table 4, Figure 6 and spatial distribution Figure S4). Total a posteriori 442 emissions for the six European regions reached a maximum of  $0.30 \pm 0.05$  Gg yr<sup>-1</sup> in 2013 443 declining to  $0.17 \pm 0.03$  Gg yr<sup>-1</sup> in 2016 and showed a slightly negative, statistically insignificant 444 trend over the period analysed (2009-2016). The cumulative European HFC-23 emissions from 445 446 2010-2016 were ~1.3 Gg corresponding to just 1.5% of our cumulative global HFC-23 emissions over this same period of 89 Gg (Table 2). Considerable differences between the two 447 inversions with different *a priori* emission distributions (see section 2.7.3) were observed on a 448 country scale, with generally larger Italian a posteriori emissions when the original UNFCCC 449 450 split of point and area sources was used in the *a priori* (UNFCCC\_org). In this case the inversion was not able to completely relocate the area emissions, but at the same time increased emissions
at the point source locations and resulting in overall larger *a posteriori* emissions. For both
sensitivity inversions the fraction of European emissions within grid boxes containing HCFC-22
production facilities, increased in the *a posteriori* as compared with the *a priori* distribution
(Table 4 and Figure S4).

In the following section we attempt to reconcile the changing trends in global HFC-23 emissions with the decrease in global HCFC-22 emissions after 2010 and the decline in the annual HCFC-22 growth rate. There are a number of key factors which we believe can explain the changing trend in the recent history of HFC-23 emissions after the minimum in 2009.

3.4. Factors affecting the recent increase in HFC-23 emissions and changes in the461 consumption of produced HCFC-22.

Recent publications have highlighted the substantial increase in HCFC-22 production in 462 1. Non-Annex 1 countries since the 1990s, especially in the last decade, due to increasing demand 463 in air-conditioning, refrigeration applications and primarily from the use of HCFC-22 as a 464 feedstock in fluoropolymer manufacture (UNEP, 2009; Miller et al., 2010; USEPA, 2013; Fang 465 466 et al., 2015). This has resulted in Non-Annex 1 countries emitting more co-produced HFC-23 than Annex 1 countries since about 2001 (Miller et al., 2010). HCFC-22 production in Annex 1 467 countries in 2015 had shrunk by 45% from the peak historic value of 407 Gg yr<sup>-1</sup> in 1996 (see 468 Supplementary Material (3), Table S3). This has been accomplished by plant closures and 469 further reductions of HFC-23 emissions by enhanced destruction in the remaining plants. 470 Nevertheless, ~ 1 Gg of HFC-23 was emitted in 2015 from HCFC-22 production in Annex 1 471 472 countries, mainly from Russia and USA (96%). For comparison, the combined HFC-23 emissions in 2015 from the six European regions (listed in Table 4) were just  $0.11 \pm 0.03$  Gg. 473 474

2. Since 2006, a major factor mitigating HFC-23 emissions has been the CERs issued under 475 the CDM. However, under the original CDM rules, large CERs that cost relatively little to 476 acquire could be claimed legitimately (Munnings et al., 2016) and the rules were changed to bar 477 new entrants to the mechanism after 2009. The HFC-23/HCFC-22 co-production or waste gas 478 generation ratio varies between 1.5 - 4% (by mass) depending on HCFC-22 plant operating 479 conditions and process optimization (McCulloch and Lindley, 2007). Under the 480 UNFCCC/CDM, 19 HCFC-22 production plants in five Non-Annex 1 countries - Argentina (1), 481 China (11), Democratic People's Republic of Korea (1), India (5) and Mexico (1) - were 482 approved for participation in CDM projects. These countries reportedly incinerated 5.7 Gg and 483 6.8 Gg of HFC-23 in 2007 and 2008, respectively (UNFCCC, 2009). This represented 43 - 48% 484

of the HCFC-22 produced in Non-Annex 1 countries during 2007-2008 assuming the 1.5 - 4% 485 486 co-production factor (Montzka et al., 2010). These five countries produced 597 Gg of HCFC-22 from controlled and feed stock uses in 2015. HFC-23 generated from this HCFC-22 production 487 was estimated at 16 Gg with an average co-production ratio of 2.6%. Furthermore, it was 488 estimated that China produced 535 Gg of HCFC-22 in 2015 (~90% of the five countries' total 489 production) and 45% of the co-produced HFC-23 generated was destroyed in the CDM 490 destruction facilities (UNEP, 2017a). The first seven-year crediting period of CDM projects in 491 China expired in 2013, concurrent with the European Union ceasing the purchase of CER credits 492 for HFC-23 produced in industrial processes after May 2013 (Fang et al., 2014). 493

3. Lastly, we should consider whether there are other sources of HFC-23 which might 496 explain an increase in global emissions. While the major source of all HFC-23 is HCFC-22 coproduction, material that is recovered and sold may subsequently be emitted to the atmosphere. 497 Emissions of HFC-23 from fire suppression systems are negligible relative to global production 498 (McCulloch and Lindley, 2007) and emissions from all emissive uses are reported to be  $0.13 \pm$ 499 0.01 Gg yr<sup>-1</sup> from Annex 1 countries (UNFCCC, 2017) and less than 0.003 Gg for the 500 refrigeration and fire-fighting sectors in 2015 in the five Non-Annex 1 countries listed above, 501 502 (UNEP, 2017a). Semi-conductor use of HFC-23 is insignificant having been replaced by more efficient etchants and where destruction efficiencies are greater than 90% (Bartos et al., 2006, 503 504 Miller et al., 2010). Fraser et al. (2013) reported a very small emissions factor of 0.04 g HFC-23 Mg<sup>-1</sup> aluminium (Al) from the Kurri Kurri smelter in NSW, Australia. It was estimated that this 505 corresponds to an annual emission of HFC-23 from Al production of ~0.003 Gg based on a 506 global Al production of 57 Tg in 2016 (http://www.world-aluminium.org/statistics/#data, 507 accessed 2016). Realistically, these other potential industrial sources of HFC-23 emissions are 508 very small ( $< 0.015 \text{ Gg yr}^{-1}$ ) in the context of global emissions estimates. 509

The combination of these factors strongly suggests that the steep reversal of the downwards trend in HFC-23 emissions after 2009 is attributable to HFC-23 abatement measures not being adequate to offset the increasing growth in production of HCFC-22 for non-dispersive feedstock. This is despite the initial success of CDM abatement technologies leading to mitigation of HFC-23 emissions during 2006-2009.

### 515 **4.** Conclusions

The introduction of CERs under the CDM did contribute to a reduction of HFC-23 emissions in Non-Annex 1 countries during 2006-2009, thereby lowering global emissions, reaching a minimum of  $9.6 \pm 0.6$  Gg yr<sup>-1</sup> in 2009. However, from 2010 to 2014 global HFC-23

- emissions increased steadily at an annual average rate of  $\sim 1$  Gg yr<sup>-1</sup> reaching a new maximum of 519  $14.5 \pm 0.6$  Gg yr<sup>-1</sup> in 2014. This period coincides with the highest levels of our bottom-up 520 estimates of HCFC-22 production in Non-Annex 1 countries (Supplementary Material (3), Table 521 S3), coinciding with a transition period when HCFC-22 production plants without any abatement 522 controls had yet to install incineration technologies or fully adopt process optimisation 523 techniques. Furthermore, non-CDM plants are not required to report co-produced HFC-23 524 emissions, although Fang et al. (2015) calculated that these plants have a lower HFC-23/HCFC-525 22 production ratio as they came into operation after the CDM period and would most likely be 526 using improved technologies for HFC-23 abatement. 527
- Our cumulative HFC-23 emissions estimates from 2010 to 2016 were  $89.1 \pm 4.3$  Gg  $(1.1 \pm 0.2)$
- 529 Gt CO<sub>2</sub>-eq), which led to an increase in radiative forcing of  $1.0 \pm 0.1$  mW m<sup>-2</sup>. This implies that
- the post-2009 increase in HFC-23 emissions resulted from the decision not to award new CDM
- projects after 2009, against a background of increasing production of HCFC-22 in plants that did not have abatement technology. Over this same time frame, the magnitude of the cumulative emissions of HCFC-22 was  $2610 \pm 311$  Gg (4724 Mt CO<sub>2</sub>-eq yr<sup>-1</sup>). During 2015-2016 our results show a decline of about 9% in average global HFC-23 emissions (12.9 Gg yr<sup>-1</sup>) relative to the
- 2013-2014 average of 14.2 Gg yr<sup>-1</sup>. We note that in the Kigali 2016 Amendment to the Montreal
- Protocol, China committed to a domestic dispersive HCFC-22 production reduction of 10% by
- 2015 compared to the average 2009-2010 production (UNEP, 2017a). While this regulation
- should decrease HCFC-22 production for dispersive uses, overall HFC-23 emissions could still
- continue to increase due to a potential increase in the production of HCFC-22 for feedstock uses
- (Fang et al., 2014). It is perhaps encouraging that the Kigali Amendment also stipulates that the
- Parties to the Montreal Protocol shall ensure that HFC-23 emissions generated from production
- facilities producing HCFCs or HFCs are destroyed to the maximum extent possible using
- technology yet to be approved by the Parties. In 2014, with the support of the Chinese
- Government, 13 new destruction facilities at 15 HCFC-22 production lines not covered by
- CDM, were installed (UNEP, 2017a). The time frame of these new initiatives is consistent with
  the most recent reduction (2015-2016) of global HFC-23 emissions.
- The mismatch between mitigation and emissions, that is most evident in China, suggests that the delay in the implementation of additional abatement measures allowed HFC-23 emissions to increase before these measures became effective. Our results imply that HFC-23 emissions into the atmosphere will continue to increase and make a contribution to radiative forcing of HFCs until the implementation of abatement becomes a universal requirement.

### 554 Acknowledgements

We specifically acknowledge the cooperation and efforts of the station operators (G. 555 Spain, Mace Head, Ireland; R. Dickau, Trinidad Head, California; P. Sealy, Ragged Point, 556 Barbados; NOAA Officer-in-Charge, Cape Matatula, American Samoa; S. Cleland, Cape Grim, 557 Tasmania) and their staff at all the AGAGE stations. The operation of the AGAGE stations was 558 supported by the National Aeronautics and Space Administration (NASA, USA) (grants NAG5-559 12669, NNX07AE89G, NNX11AF17G and NNX16AC98G to MIT; grants NAG5-4023, 560 NNX07AE87G, NNX07AF09G, NNX11AF15G, NNX11AF16G, NNX16AC96G and 561 NNX16A97G to SIO). We acknowledge the Department for Business, Energy and Industrial 562 Strategy (BEIS, UK) for contract 1028/06/2015 to the University of Bristol and the UK 563 564 Meteorological Office in support of Mace Head, Ireland, and modelling activities; the National Oceanic and Atmospheric Administration (NOAA, USA), contract RA-133-R15-CN-0008 to the 565 University of Bristol in support of Ragged Point, Barbados; we acknowledge NOAA support for 566 the operations of the American Samoa station, and support by the Commonwealth Scientific and 567 Industrial Research Organisation (CSIRO, Australia), the Bureau of Meteorology (Australia) and 568 Refrigerant Reclaim Australia for Cape Grim operations. 569

CSIRO's contribution was supported in part by the Australian Climate Change Science 570 Program (ACCSP), an Australian Government Initiative. Australian firn activities in the 571 Antarctic are specifically supported by the Australian Antarctic Science Program. We 572 573 acknowledge the members of the firn air sampling teams for provision of the samples from Law Dome, NEEM, and South Pole. NEEM is directed and organized by the Centre of Ice and 574 Climate at the Niels Bohr Institute and US NSF, Office of Polar Programs. It is supported by 575 funding agencies and institutions in Belgium (FNRS-CFB and FWO), Canada (NRCan/GSC), 576 China (CAS), Denmark (FIST), France (IPEV, CNRS/INSU, CEA, and ANR), Germany (AWI), 577 Iceland (RannIs), Japan (NIPR), Korea (KOPRI), the Netherlands (NWO/ALW), Sweden (VR), 578 Switzerland (SNF), United Kingdom (NERC), and the U.S. (U.S. NSF, Office of Polar 579 Programs). Financial support for the Jungfraujoch measurements is acknowledged from the 580 Swiss national programme HALCLIM (Swiss Federal Office for the Environment (FOEN)). 581 Support for the Jungfraujoch station was provided by International Foundation High Altitude 582 Research Stations Jungfraujoch and Gornergrat (HFSJG). 583 M. Rigby is supported by a NERC Advanced Fellowship NE/I021365/1. We acknowledge the 584 cooperation of R. Langenfelds for his long-term involvement in supporting and maintaining the 585

Cape Grim Air Archive. We also thank S. Montzka and B. Hall for supplying actual datasets

from their publications.

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

### 838 Data availability

- The entire ALE/GAGE/AGAGE data base comprising every calibrated measurement including pollution
- events is archived with the Carbon Dioxide Information and Analysis Center (CDIAC) at the U.S.
- Department of Energy, Oak Ridge National Laboratory (<u>http://cdiac.ornl.gov</u> and also
- (<u>http://agage.mit.edu/data/agage-data)</u>.

Table 1. AGAGE sites used in this study, their coordinates and start dates for GC-MS-Medusa
measurements of HFC-23 and HCFC-22.

| AGAGE Site                             | Latitude | Longitude | HFC-23    | HCFC-22   |
|----------------------------------------|----------|-----------|-----------|-----------|
|                                        |          |           |           |           |
| Mace Head (MHD), Ireland <sup>1</sup>  | 53.3° N  | 9.9° W    | Oct. 2007 | Nov. 2003 |
| Trinidad Head (THD), California, USA   | 41.0° N  | 124.1° W  | Sep. 2007 | Mar. 2005 |
| Ragged Point (RPB), Barbados           | 13.2° N  | 59.4° W   | Aug. 2007 | May. 2005 |
| Cape Matatula (SMO), American Samoa    | 14.2° S  | 170.6° W  | Oct. 2007 | May. 2006 |
| Cape Grim (CGO), Tasmania, Australia   | 40.7° S  | 144.7° E  | Nov. 2007 | Jan 2004  |
| Jungfraujoch, Switzerland <sup>1</sup> | 46.5°N   | 8.0°E     | Apr. 2008 | Aug. 2012 |
| Cape Grim Air Archive                  |          |           | Apr. 1978 | Apr. 1978 |

Observations used for regional European emissions

- 850
- 050

Table 2. Annual mean global HFC-23 emissions, mole fractions, and growth rates, derived from

the AGAGE 12-box model.

| Year | HFC-23 global annual             | HFC-23 global mean mole                     | HFC-23 global growth rate      |
|------|----------------------------------|---------------------------------------------|--------------------------------|
|      | emissions (Gg yr <sup>-1</sup> ) | fraction (pmol $mol^{-1}$ )                 | $(pmol mol^{-1} yr^{-1})$      |
|      | $\pm 1$ sigma ( $\sigma$ ) SD.   | $\pm 1 \text{ sigma } (\sigma) \text{ SD.}$ | $\pm 1$ sigma ( $\sigma$ ) SD. |
| 1980 | $4.2 \pm 0.7$                    | $3.9 \pm 0.1$                               | $0.33 \pm 0.05$                |
| 1981 | $4.2\pm0.8$                      | $4.3 \pm 0.1$                               | $0.33 \pm 0.05$                |
| 1982 | $4.4\pm0.7$                      | $4.6 \pm 0.1$                               | $0.35\pm0.07$                  |
| 1983 | $5.2\pm0.7$                      | $5.0\pm0.1$                                 | $0.41 \pm 0.05$                |
| 1984 | $5.6 \pm 0.7$                    | $5.4 \pm 0.1$                               | $0.44 \pm 0.05$                |
| 1985 | $5.8\pm0.8$                      | $5.9\pm0.1$                                 | $0.45 \pm 0.05$                |
| 1986 | $5.8\pm0.7$                      | $6.3 \pm 0.1$                               | $0.45\pm0.05$                  |
| 1987 | $5.9\pm0.7$                      | $6.8 \pm 0.1$                               | $0.46 \pm 0.05$                |
| 1988 | $6.8\pm0.7$                      | $7.2\pm0.1$                                 | $0.52 \pm 0.05$                |
| 1989 | $7.1\pm0.7$                      | $7.8\pm0.2$                                 | $0.55\pm0.05$                  |
| 1990 | $7.0\pm0.7$                      | $8.3 \pm 0.2$                               | $0.54 \pm 0.05$                |
| 1991 | $7.0\pm0.7$                      | $8.9\pm0.2$                                 | $0.54 \pm 0.05$                |
| 1992 | $7.4 \pm 0.6$                    | $9.4 \pm 0.2$                               | $0.57\pm0.05$                  |
| 1993 | $7.9\pm0.6$                      | $10.0 \pm 0.2$                              | $0.61 \pm 0.04$                |
| 1994 | $8.3 \pm 0.7$                    | $10.6 \pm 0.2$                              | $0.64 \pm 0.04$                |
| 1995 | $8.9\pm0.6$                      | $11.3 \pm 0.2$                              | $0.69 \pm 0.05$                |
| 1996 | $9.6 \pm 0.6$                    | $12.0 \pm 0.2$                              | $0.74\pm0.04$                  |
| 1997 | $10.1 \pm 0.6$                   | $12.8 \pm 0.3$                              | $0.77\pm0.04$                  |
| 1998 | $10.4 \pm 0.7$                   | $13.6\pm0.3$                                | $0.79 \pm 0.04$                |
| 1999 | $10.9\pm0.7$                     | $14.4 \pm 0.3$                              | $0.82 \pm 0.04$                |
| 2000 | $10.4 \pm 0.8$                   | $15.2 \pm 0.3$                              | $0.76\pm0.05$                  |
| 2001 | $9.4 \pm 0.7$                    | $15.9 \pm 0.3$                              | $0.68 \pm 0.05$                |
| 2002 | $9.5 \pm 0.7$                    | $16.6 \pm 0.3$                              | $0.69 \pm 0.05$                |
| 2003 | $10.3 \pm 0.8$                   | $17.3 \pm 0.3$                              | $0.77\pm0.05$                  |
| 2004 | $11.8 \pm 0.8$                   | $18.1 \pm 0.3$                              | $0.90 \pm 0.05$                |
| 2005 | $13.2 \pm 0.8$                   | $19.1 \pm 0.4$                              | $1.01 \pm 0.05$                |
| 2006 | $13.3\pm0.8$                     | $20.1 \pm 0.4$                              | $0.99 \pm 0.05$                |
| 2007 | $11.7 \pm 0.7$                   | $21.0 \pm 0.4$                              | $0.85 \pm 0.04$                |
| 2008 | $11.2 \pm 0.6$                   | $21.9\pm0.4$                                | $0.78\pm0.03$                  |
| 2009 | $9.6 \pm 0.6$                    | $22.6 \pm 0.4$                              | $0.68 \pm 0.03$                |
| 2010 | $10.4 \pm 0.6$                   | $23.3 \pm 0.4$                              | $0.74 \pm 0.03$                |
| 2011 | $11.6 \pm 0.6$                   | $24.1 \pm 0.5$                              | $0.85 \pm 0.03$                |
| 2012 | $12.9\pm0.6$                     | $25.0\pm0.5$                                | $0.96\pm0.03$                  |
| 2013 | $14.0 \pm 0.6$                   | $26.0\pm0.5$                                | $1.04 \pm 0.03$                |
| 2014 | $14.5 \pm 0.6$                   | $27.0\pm0.5$                                | $1.05 \pm 0.03$                |
| 2015 | $13.1 \pm 0.7$                   | $28.0\pm0.5$                                | $0.95 \pm 0.03$                |
| 2016 | $12.7\pm0.6$                     | $28.9\pm0.6$                                | $0.94 \pm 0.03$                |
|      | Note: Data are tabulated as      | annual mean mid-year values.                |                                |
|      | Earlier emissions estimates      | (1930-1979) determined from                 | the AGAGE 12-box model are     |

listed in Supplementary Material (6), Table S5.

855

from the AGAGE 12-box model.

| Year | HCFC-22 global annual                         | HCFC-22 global mean                       | HCFC-22 global growth                           |
|------|-----------------------------------------------|-------------------------------------------|-------------------------------------------------|
|      | emissions (Gg vr <sup><math>-1</math></sup> ) | mole fraction (pmol mol <sup>-1</sup> )   | rate (pmol mol <sup>-1</sup> yr <sup>-1</sup> ) |
|      | $\pm 1$ sigma ( $\sigma$ ) SD.                | $\pm 1 \text{ sigma}(\sigma) \text{ SD}.$ | $\pm 1$ sigma ( $\sigma$ ) SD.                  |
| 1980 | $116.8 \pm 17.6$                              | $41.6 \pm 1.0$                            | $4.1 \pm 0.9$                                   |
| 1981 | $123.4 \pm 18.8$                              | $45.8 \pm 1.2$                            | $4.2 \pm 0.8$                                   |
| 1982 | $125.7 \pm 15.7$                              | $49.9 \pm 1.3$                            | $4.0 \pm 0.8$                                   |
| 1983 | $125.2 \pm 20.1$                              | $53.7 \pm 0.9$                            | $3.6 \pm 1.0$                                   |
| 1984 | $136.8 \pm 18.1$                              | $57.3 \pm 1.1$                            | $4.2 \pm 0.6$                                   |
| 1985 | $166.1 \pm 17.0$                              | $62.2 \pm 1.0$                            | $5.7\pm0.7$                                     |
| 1986 | $174.2 \pm 19.3$                              | $68.5 \pm 1.0$                            | $5.5 \pm 0.7$                                   |
| 1987 | $155.4 \pm 20.2$                              | $72.9 \pm 1.2$                            | $4.1 \pm 0.7$                                   |
| 1988 | $177.8 \pm 19.6$                              | $77.2 \pm 1.0$                            | $5.0 \pm 0.7$                                   |
| 1989 | $198.0\pm19.8$                                | $82.9\pm1.0$                              | $6.0 \pm 0.7$                                   |
| 1990 | $209.1\pm19.7$                                | $89.1 \pm 1.2$                            | $6.2 \pm 0.6$                                   |
| 1991 | $212.2 \pm 21.4$                              | $95.1 \pm 1.2$                            | $5.8\pm0.7$                                     |
| 1992 | $207.1 \pm 23.9$                              | $100.5 \pm 1.3$                           | $5.0\pm0.6$                                     |
| 1993 | $214.6\pm22.8$                                | $105.4 \pm 1.3$                           | $5.0\pm0.6$                                     |
| 1994 | $222.7\pm22.9$                                | $110.4 \pm 1.3$                           | $5.2\pm0.5$                                     |
| 1995 | $241.4\pm26.9$                                | $115.9 \pm 1.3$                           | $5.7\pm0.5$                                     |
| 1996 | $230.1 \pm 24.3$                              | $121.4 \pm 1.3$                           | $4.8\pm0.5$                                     |
| 1997 | $238.3 \pm 24.1$                              | $125.8 \pm 1.3$                           | $5.0\pm0.5$                                     |
| 1998 | $256.0\pm27.2$                                | $131.7 \pm 1.3$                           | $5.7\pm0.4$                                     |
| 1999 | $251.8\pm28.4$                                | $136.8 \pm 1.4$                           | $5.0 \pm 0.3$                                   |
| 2000 | $275.1 \pm 28.7$                              | $142.0 \pm 1.4$                           | $5.8 \pm 0.2$                                   |
| 2001 | $275.3 \pm 28.8$                              | $147.9 \pm 1.5$                           | $5.5 \pm 0.2$                                   |
| 2002 | $280.7\pm31.6$                                | $153.1 \pm 1.6$                           | $5.3 \pm 0.2$                                   |
| 2003 | $286.3 \pm 30.1$                              | $158.3 \pm 1.6$                           | $5.3 \pm 0.2$                                   |
| 2004 | $292.1 \pm 30.6$                              | $163.7 \pm 1.7$                           | $5.3 \pm 0.2$                                   |
| 2005 | $312.3 \pm 34.6$                              | $169.2 \pm 1.6$                           | $6.1 \pm 0.2$                                   |
| 2006 | $334.3 \pm 35.0$                              | $175.9 \pm 1.7$                           | $7.2 \pm 0.2$                                   |
| 2007 | $355.6 \pm 35.2$                              | $183.6 \pm 1.8$                           | $8.0 \pm 0.2$                                   |
| 2008 | $372.9 \pm 38.4$                              | $191.9\pm1.9$                             | $7.9 \pm 0.2$                                   |
| 2009 | $368.6 \pm 39.7$                              | $199.2 \pm 2.0$                           | $7.4 \pm 0.2$                                   |
| 2010 | $385.8 \pm 41.3$                              | $206.8 \pm 2.0$                           | $7.4 \pm 0.3$                                   |
| 2011 | $373.1 \pm 41.3$                              | $213.7 \pm 2.1$                           | $6.2 \pm 0.2$                                   |
| 2012 | $373.2 \pm 45.5$                              | $219.3 \pm 2.2$                           | $5.5 \pm 0.2$                                   |
| 2013 | $369.6 \pm 44.1$                              | $224.7 \pm 2.3$                           | $5.0 \pm 0.2$                                   |
| 2014 | $373.9 \pm 45.7$                              | $229.5 \pm 2.3$                           | $4.6 \pm 0.2$                                   |
| 2015 | $364.2 \pm 47.7$                              | $233.7 \pm 2.2$                           | $3.9 \pm 0.2$                                   |
| 2016 | $370.3 \pm 45.9$                              | $237.5 \pm 2.2$                           | $3.9 \pm 0.2$                                   |

Note: Data are tabulated as annual mean mid-year values. These HCFC-22 global emissions
estimates are updates of the HCFC-22 emissions reported in Simmonds et al. (2017) for the
period 1995-2015.

Table 4: European HFC-23 emissions (tonne, Mg) by country/region: E<sub>a</sub> *a priori*, E<sub>b</sub> *a posteriori*emissions, f<sub>a</sub> fraction of *a priori* emissions from factory locations, f<sub>b</sub> fraction of *a posteriori*emissions from factory locations. All values represent averages from both inversions using
different *a priori* distributions.

| ermany         |                                                                                                                                                                                                                                                            |                                                                                                                                                                                                                                                                                                                                                                             | France                                                                                                                                                                                                                                                                                                                                                                                                                                                                                                                     |                                                                                                                                                                                                                                                                                                                                                                                                                  |                                                                                                                                                                                                                                                                                                                                                                                                                                                                                                                                                                                                                                                                                                                                                                    |                                                                                                                                                                                                                                                                                                                                                                                                                                                                                                                                    | Italy                                                                                                                                                                                                                                                                                                                                                                                                                                                                                                                                                                                                                                      |                                                                                                                                                                                                                                                                                                                                                                                                                                                                                                                                                                                                                                                                                                                                                                                                                                        |                                                                                                                                                                                                                                                                                                                                                                                             |
|----------------|------------------------------------------------------------------------------------------------------------------------------------------------------------------------------------------------------------------------------------------------------------|-----------------------------------------------------------------------------------------------------------------------------------------------------------------------------------------------------------------------------------------------------------------------------------------------------------------------------------------------------------------------------|----------------------------------------------------------------------------------------------------------------------------------------------------------------------------------------------------------------------------------------------------------------------------------------------------------------------------------------------------------------------------------------------------------------------------------------------------------------------------------------------------------------------------|------------------------------------------------------------------------------------------------------------------------------------------------------------------------------------------------------------------------------------------------------------------------------------------------------------------------------------------------------------------------------------------------------------------|--------------------------------------------------------------------------------------------------------------------------------------------------------------------------------------------------------------------------------------------------------------------------------------------------------------------------------------------------------------------------------------------------------------------------------------------------------------------------------------------------------------------------------------------------------------------------------------------------------------------------------------------------------------------------------------------------------------------------------------------------------------------|------------------------------------------------------------------------------------------------------------------------------------------------------------------------------------------------------------------------------------------------------------------------------------------------------------------------------------------------------------------------------------------------------------------------------------------------------------------------------------------------------------------------------------|--------------------------------------------------------------------------------------------------------------------------------------------------------------------------------------------------------------------------------------------------------------------------------------------------------------------------------------------------------------------------------------------------------------------------------------------------------------------------------------------------------------------------------------------------------------------------------------------------------------------------------------------|----------------------------------------------------------------------------------------------------------------------------------------------------------------------------------------------------------------------------------------------------------------------------------------------------------------------------------------------------------------------------------------------------------------------------------------------------------------------------------------------------------------------------------------------------------------------------------------------------------------------------------------------------------------------------------------------------------------------------------------------------------------------------------------------------------------------------------------|---------------------------------------------------------------------------------------------------------------------------------------------------------------------------------------------------------------------------------------------------------------------------------------------------------------------------------------------------------------------------------------------|
| f <sub>a</sub> | $f_b$                                                                                                                                                                                                                                                      | Ea                                                                                                                                                                                                                                                                                                                                                                          | Eb                                                                                                                                                                                                                                                                                                                                                                                                                                                                                                                         | fa                                                                                                                                                                                                                                                                                                                                                                                                               | $f_b$                                                                                                                                                                                                                                                                                                                                                                                                                                                                                                                                                                                                                                                                                                                                                              | Ea                                                                                                                                                                                                                                                                                                                                                                                                                                                                                                                                 | E <sub>b</sub>                                                                                                                                                                                                                                                                                                                                                                                                                                                                                                                                                                                                                             | fa                                                                                                                                                                                                                                                                                                                                                                                                                                                                                                                                                                                                                                                                                                                                                                                                                                     | $f_b$                                                                                                                                                                                                                                                                                                                                                                                       |
| lg/yr) (%)     | (%)                                                                                                                                                                                                                                                        | (Mg/yr)                                                                                                                                                                                                                                                                                                                                                                     | (Mg/yr)                                                                                                                                                                                                                                                                                                                                                                                                                                                                                                                    | (%)                                                                                                                                                                                                                                                                                                                                                                                                              | (%)                                                                                                                                                                                                                                                                                                                                                                                                                                                                                                                                                                                                                                                                                                                                                                | (Mg/yr)                                                                                                                                                                                                                                                                                                                                                                                                                                                                                                                            | (Mg/yr)                                                                                                                                                                                                                                                                                                                                                                                                                                                                                                                                                                                                                                    | (%)                                                                                                                                                                                                                                                                                                                                                                                                                                                                                                                                                                                                                                                                                                                                                                                                                                    | (%)                                                                                                                                                                                                                                                                                                                                                                                         |
| 34±12 30       | 49                                                                                                                                                                                                                                                         | 15±31                                                                                                                                                                                                                                                                                                                                                                       | 2±3.2                                                                                                                                                                                                                                                                                                                                                                                                                                                                                                                      | 88                                                                                                                                                                                                                                                                                                                                                                                                               | 6                                                                                                                                                                                                                                                                                                                                                                                                                                                                                                                                                                                                                                                                                                                                                                  | 8.4±17                                                                                                                                                                                                                                                                                                                                                                                                                                                                                                                             | 34±8.7                                                                                                                                                                                                                                                                                                                                                                                                                                                                                                                                                                                                                                     | 26                                                                                                                                                                                                                                                                                                                                                                                                                                                                                                                                                                                                                                                                                                                                                                                                                                     | 52                                                                                                                                                                                                                                                                                                                                                                                          |
| 19±14 27       | 13                                                                                                                                                                                                                                                         | 12±23                                                                                                                                                                                                                                                                                                                                                                       | 15±5.1                                                                                                                                                                                                                                                                                                                                                                                                                                                                                                                     | 84                                                                                                                                                                                                                                                                                                                                                                                                               | 86                                                                                                                                                                                                                                                                                                                                                                                                                                                                                                                                                                                                                                                                                                                                                                 | 9±18                                                                                                                                                                                                                                                                                                                                                                                                                                                                                                                               | 48±9.3                                                                                                                                                                                                                                                                                                                                                                                                                                                                                                                                                                                                                                     | 26                                                                                                                                                                                                                                                                                                                                                                                                                                                                                                                                                                                                                                                                                                                                                                                                                                     | 45                                                                                                                                                                                                                                                                                                                                                                                          |
| 44±13 33       | 58                                                                                                                                                                                                                                                         | 7.7±15                                                                                                                                                                                                                                                                                                                                                                      | 10±3.4                                                                                                                                                                                                                                                                                                                                                                                                                                                                                                                     | 70                                                                                                                                                                                                                                                                                                                                                                                                               | 73                                                                                                                                                                                                                                                                                                                                                                                                                                                                                                                                                                                                                                                                                                                                                                 | 9.2±18                                                                                                                                                                                                                                                                                                                                                                                                                                                                                                                             | 34±11                                                                                                                                                                                                                                                                                                                                                                                                                                                                                                                                                                                                                                      | 26                                                                                                                                                                                                                                                                                                                                                                                                                                                                                                                                                                                                                                                                                                                                                                                                                                     | 43                                                                                                                                                                                                                                                                                                                                                                                          |
| 32±9.7 30      | 40                                                                                                                                                                                                                                                         | 8.1±16                                                                                                                                                                                                                                                                                                                                                                      | 8.3±3.4                                                                                                                                                                                                                                                                                                                                                                                                                                                                                                                    | 76                                                                                                                                                                                                                                                                                                                                                                                                               | 73                                                                                                                                                                                                                                                                                                                                                                                                                                                                                                                                                                                                                                                                                                                                                                 | 9.1±18                                                                                                                                                                                                                                                                                                                                                                                                                                                                                                                             | 25±8.6                                                                                                                                                                                                                                                                                                                                                                                                                                                                                                                                                                                                                                     | 26                                                                                                                                                                                                                                                                                                                                                                                                                                                                                                                                                                                                                                                                                                                                                                                                                                     | 31                                                                                                                                                                                                                                                                                                                                                                                          |
| 16±9.4 28      | 58                                                                                                                                                                                                                                                         | 9.2±18                                                                                                                                                                                                                                                                                                                                                                      | 27±16                                                                                                                                                                                                                                                                                                                                                                                                                                                                                                                      | 82                                                                                                                                                                                                                                                                                                                                                                                                               | 93                                                                                                                                                                                                                                                                                                                                                                                                                                                                                                                                                                                                                                                                                                                                                                 | 9.3±19                                                                                                                                                                                                                                                                                                                                                                                                                                                                                                                             | 47±24                                                                                                                                                                                                                                                                                                                                                                                                                                                                                                                                                                                                                                      | 26                                                                                                                                                                                                                                                                                                                                                                                                                                                                                                                                                                                                                                                                                                                                                                                                                                     | 29                                                                                                                                                                                                                                                                                                                                                                                          |
| 20±9.2 30      | 27                                                                                                                                                                                                                                                         | 9.4±19                                                                                                                                                                                                                                                                                                                                                                      | 10±2.3                                                                                                                                                                                                                                                                                                                                                                                                                                                                                                                     | 84                                                                                                                                                                                                                                                                                                                                                                                                               | 85                                                                                                                                                                                                                                                                                                                                                                                                                                                                                                                                                                                                                                                                                                                                                                 | 9.5±19                                                                                                                                                                                                                                                                                                                                                                                                                                                                                                                             | 32±14                                                                                                                                                                                                                                                                                                                                                                                                                                                                                                                                                                                                                                      | 26                                                                                                                                                                                                                                                                                                                                                                                                                                                                                                                                                                                                                                                                                                                                                                                                                                     | 32                                                                                                                                                                                                                                                                                                                                                                                          |
| 12±8.7 29      | 14                                                                                                                                                                                                                                                         | 9.5±19                                                                                                                                                                                                                                                                                                                                                                      | 17±3.9                                                                                                                                                                                                                                                                                                                                                                                                                                                                                                                     | 86                                                                                                                                                                                                                                                                                                                                                                                                               | 92                                                                                                                                                                                                                                                                                                                                                                                                                                                                                                                                                                                                                                                                                                                                                                 | 9.8±20                                                                                                                                                                                                                                                                                                                                                                                                                                                                                                                             | 37±19                                                                                                                                                                                                                                                                                                                                                                                                                                                                                                                                                                                                                                      | 26                                                                                                                                                                                                                                                                                                                                                                                                                                                                                                                                                                                                                                                                                                                                                                                                                                     | 24                                                                                                                                                                                                                                                                                                                                                                                          |
| 19±9.1 29      | 26                                                                                                                                                                                                                                                         | 9.5±19                                                                                                                                                                                                                                                                                                                                                                      | 9.9±3.4                                                                                                                                                                                                                                                                                                                                                                                                                                                                                                                    | 86                                                                                                                                                                                                                                                                                                                                                                                                               | 85                                                                                                                                                                                                                                                                                                                                                                                                                                                                                                                                                                                                                                                                                                                                                                 | 9.8±20                                                                                                                                                                                                                                                                                                                                                                                                                                                                                                                             | 23±10                                                                                                                                                                                                                                                                                                                                                                                                                                                                                                                                                                                                                                      | 26                                                                                                                                                                                                                                                                                                                                                                                                                                                                                                                                                                                                                                                                                                                                                                                                                                     | 39                                                                                                                                                                                                                                                                                                                                                                                          |
|                | fa           g/yr)         (%)           34±12         30           19±14         27           44±13         33           32±9.7         30           6±9.4         28           20±9.2         30           12±8.7         29           19±9.1         29 | fa         fb           g/yr)         (%)         (%)           34±12         30         49           19±14         27         13           44±13         33         58           32±9.7         30         40           .6±9.4         28         58           20±9.2         30         27           .12±8.7         29         14           .9±9.1         29         26 | fa         fb         Ea           g/yr)         (%)         (%)         (Mg/yr)           34±12         30         49         15±31           19±14         27         13         12±23           44±13         33         58         7.7±15           82±9.7         30         40         8.1±16           6±9.4         28         58         9.2±18           20±9.2         30         27         9.4±19           12±8.7         29         14         9.5±19           19±9.1         29         26         9.5±19 | rmany $F_a$ $F_b$ $E_a$ $F_b$ $g/yr$ $(\%)$ $(Mg/yr)$ $(Mg/yr)$ $34\pm12$ $30$ $49$ $15\pm31$ $2\pm3.2$ $19\pm14$ $27$ $13$ $12\pm23$ $15\pm5.1$ $44\pm13$ $33$ $58$ $7.7\pm15$ $10\pm3.4$ $42\pm9.7$ $30$ $40$ $8.1\pm16$ $8.3\pm3.4$ $6\pm9.4$ $28$ $58$ $9.2\pm18$ $27\pm16$ $20\pm9.2$ $30$ $27$ $9.4\pm19$ $10\pm2.3$ $22\pm8.7$ $29$ $14$ $9.5\pm19$ $17\pm3.9$ $9\pm9.1$ $29$ $26$ $9.5\pm19$ $9.9\pm3.4$ | $f_a$ $f_b$ $E_a$ $E_b$ $f_a$ g/yr)         (%)         (%)         (Mg/yr)         (Mg/yr)         (%)           34±12         30         49         15±31         2±3.2         88           19±14         27         13         12±23         15±5.1         84           44±13         33         58         7.7±15         10±3.4         70           82±9.7         30         40         8.1±16         8.3±3.4         76           6±9.4         28         58         9.2±18         27±16         82           20±9.2         30         27         9.4±19         10±2.3         84           42±8.7         29         14         9.5±19         17±3.9         86           9.9±1.1         29         26         9.5±19         9.9±3.4         86 | rmany $F_a$ $F_b$ $E_a$ $F_b$ $f_a$ $f_b$ $g/yr$ $(\%)$ $(\%)$ $(Mg/yr)$ $(Mg/yr)$ $(\%)$ $(\%)$ $34\pm12$ $30$ $49$ $15\pm31$ $2\pm3.2$ $88$ $6$ $19\pm14$ $27$ $13$ $12\pm23$ $15\pm5.1$ $84$ $86$ $44\pm13$ $33$ $58$ $7.7\pm15$ $10\pm3.4$ $70$ $73$ $82\pm9.7$ $30$ $40$ $8.1\pm16$ $8.3\pm3.4$ $76$ $73$ $6\pm9.4$ $28$ $58$ $9.2\pm18$ $27\pm16$ $82$ $93$ $20\pm9.2$ $30$ $27$ $9.4\pm19$ $10\pm2.3$ $84$ $85$ $42\pm8.7$ $29$ $14$ $9.5\pm19$ $17\pm3.9$ $86$ $92$ $9.9\pm1.1$ $29$ $26$ $9.5\pm19$ $9.9\pm3.4$ $86$ $85$ | armany $F_a$ $f_b$ $E_a$ $E_b$ $f_a$ $f_b$ $E_a$ $g/yr$ $(\%)$ $(\%)$ $(Mg/yr)$ $(Mg/yr)$ $(\%)$ $(Mg/yr)$ $(Mg/yr)$ $34\pm12$ $30$ $49$ $15\pm31$ $2\pm3.2$ $88$ $6$ $8.4\pm17$ $19\pm14$ $27$ $13$ $12\pm23$ $15\pm5.1$ $84$ $86$ $9\pm18$ $44\pm13$ $33$ $58$ $7.7\pm15$ $10\pm3.4$ $70$ $73$ $9.2\pm18$ $22\pm9.7$ $30$ $40$ $8.1\pm16$ $8.3\pm3.4$ $76$ $73$ $9.1\pm18$ $6\pm9.4$ $28$ $58$ $9.2\pm18$ $27\pm16$ $82$ $93$ $9.3\pm19$ $20\pm9.2$ $30$ $27$ $9.4\pm19$ $10\pm2.3$ $84$ $85$ $9.5\pm19$ $22\pm8.7$ $29$ $14$ $9.5\pm19$ $17\pm3.9$ $86$ $92$ $9.8\pm20$ $49\pm11$ $29$ $26$ $9.5\pm19$ $9.9\pm3.4$ $86$ $85$ $9.8\pm20$ | FranceItaly $f_a$ $f_b$ $E_a$ $E_a$ $E_b$ $f_a$ $f_b$ $E_a$ $E_b$ $g/yr)$ $(\%)$ $(\%)$ $(Mg/yr)$ $(Mg/yr)$ $(\%)$ $(Mg/yr)$ $(Mg/yr)$ $(Mg/yr)$ $34\pm12$ $30$ $49$ $15\pm31$ $2\pm3.2$ $88$ $66$ $8.4\pm17$ $34\pm8.7$ $19\pm14$ $27$ $13$ $12\pm23$ $15\pm5.1$ $84$ $86$ $9\pm18$ $48\pm9.3$ $44\pm13$ $33$ $58$ $7.7\pm15$ $10\pm3.4$ $70$ $73$ $9.2\pm18$ $34\pm11$ $82\pm9.7$ $30$ $40$ $8.1\pm16$ $8.3\pm3.4$ $76$ $73$ $9.2\pm18$ $34\pm11$ $82\pm9.7$ $30$ $40$ $8.1\pm16$ $8.3\pm3.4$ $76$ $73$ $9.1\pm18$ $25\pm8.6$ $6\pm9.4$ $28$ $58$ $9.2\pm18$ $27\pm16$ $82$ $93$ $9.3\pm19$ $47\pm24$ $20\pm9.2$ $30$ $27$ $9.4\pm19$ $10\pm2.3$ $84$ $85$ $9.5\pm19$ $32\pm14$ $42\pm8.7$ $29$ $14$ $9.5\pm19$ $17\pm3.9$ $86$ $92$ $9.8\pm20$ $37\pm19$ $49\pm9.1$ $29$ $26$ $9.5\pm19$ $9.9\pm3.4$ $86$ $85$ $9.8\pm20$ $23\pm10$ | armanyfafbEaEaEbfafbEaEbfag/yr)(%)(%)(Mg/yr)(Mg/yr)(%)(%)(Mg/yr)(Mg/yr)(%)34±12304915±312±3.28868.4±1734±8.72619±14271312±2315±5.184869±1848±9.32644±1333587.7±1510±3.470739.2±1834±112642±9.730408.1±168.3±3.476739.1±1825±6.6266±9.428589.2±1827±1682939.3±1947±24266±9.429149.5±1910±2.384859.5±1932±14266±9.429149.5±1917±3.986929.8±2037±19266±9.42920209.5±199.9±3.486859.8±2023±1026 |

|      |         | Benelux        |       |       | Ui       | nited Kingd    | om    |       | Iberian Peninsula |                |                |       |  |  |
|------|---------|----------------|-------|-------|----------|----------------|-------|-------|-------------------|----------------|----------------|-------|--|--|
| year | Ea      | E <sub>b</sub> | $f_a$ | $f_b$ | Ea       | E <sub>b</sub> | $f_a$ | $f_b$ | Ea                | E <sub>b</sub> | f <sub>a</sub> | $f_b$ |  |  |
|      | (Mg/yr) | (Mg/yr)        | (%)   | (%)   | (Mg/yr)  | (Mg/yr)        | (%)   | (%)   | (Mg/yr)           | (Mg/yr)        | (%)            | (%)   |  |  |
| 2009 | 13±27   | 25±8.8         | 98    | 99    | 3.8±7.6  | 5.3±3.1        | 84    | 87    | 97±190            | 56±29          | 57             | 10    |  |  |
| 2010 | 34±67   | 16±7.1         | 99    | 98    | 1.2±2.3  | 2.8±1.9        | 48    | 71    | 130±260           | 120±33         | 66             | 52    |  |  |
| 2011 | 15±29   | 21±16          | 97    | 97    | 1±2.1    | 1.5±1.7        | 39    | 21    | 83±170            | 75±27          | 50             | 27    |  |  |
| 2012 | 11±22   | 53±13          | 96    | 99    | 0.92±1.8 | 2±1.6          | 26    | 46    | 74±150            | 46±27          | 45             | 43    |  |  |
| 2013 | 16±33   | 94±13          | 98    | 100   | 1.1±2.1  | 2.1±1.8        | 29    | 48    | 64±130            | 110±27         | 38             | 22    |  |  |
| 2014 | 3.8±7.6 | 11±6.3         | 85    | 94    | 1.3±2.5  | 6.8±2.4        | 35    | 73    | 59±120            | 55±27          | 35             | 45    |  |  |
| 2015 | 8.7±17  | 20±12          | 94    | 97    | 1.4±2.7  | 2.5±2.1        | 34    | 37    | 48±96             | 19±18          | 26             | 9     |  |  |
| 2016 | 8.7±17  | 45±11          | 94    | 99    | 1.4±2.7  | 3.8±2.2        | 34    | 42    | 48±96             | 69±24          | 26             | 33    |  |  |

Note : **Benelux** (Belgium, the Netherlands, and Luxembourg).