# Peer review of "Recent increases in the atmospheric growth rate and emissions of HFC-23 1"

_Atmospheric Chemistry and Physics, 2017_

## Referee Comment (RC1) · Anonymous Referee #1 · 10 Nov 2017

**1   Overview:**

Review of "*Recent increases in the growth rate and emissions of HFC-23 (CHF$_3$) and the link to HCFC-22 (CHClF$_2$) production*" by Simmonds *et al.*

Simmonds *et al.* present an estimate of the HFC-23 and HCFC-22 emissions using observations from the AGAGE network. They perform two Bayesian inversions: (1) in a global 12-box model and (2) a regional inversion for Europe using FLEXPART. Overall, the manuscript reaches scientifically interesting conclusions. However, many of the details necessary to follow the conclusions are not included. Specifically, most

of the details of the forward and inverse modelling are omitted. This makes it difficult to interpret the results. Additionally, some of the conclusions seem overly speculative. I would suggest major revisions for the manuscript.

**2   Major comments:**

2.1   Structure of the manuscript and description of the modelling

The authors spend most of the methods section explaining the measurement protocols and calibration methods. Are the measurements used here fundamentally different from the previous work using the same AGAGE measurements? It seems that this paper is, at it's core, an inverse modelling paper because the novel analysis is related to the derived emissions. However, the forward and inverse modelling is not well described.

The authors spend a single paragraph explaining the 12-box model. Is the box-model including a seasonal cycle or annual concentrations? If there is not a seasonal cycle, how are the authors removing the seasonal cycle from the observations? The authors mention that they use an inter-annually repeating OH but some recent work (Rigby et al., PNAS, 2017; Turner et al., PNAS, 2017) has shown variations in OH, would this be important for the modelling here?

Questions related to the 12-box model inversion: How is the inversion done? What is the state vector? Is it annual global emissions or the emissions for each of the latitudinal boxes? Presumably the distributions are Gaussian? Are there off-diagonal covariances? How are the model-data uncertainties specified (ie., what is the model error)? Some of these details could go in a supplement, but they should be described somewhere.

Similarly, the authors present a second inverse analysis within the results section (Section 3.2). The modelling is explained in a single paragraph, yet this is a very complicated inverse model they present. In the paragraph that follows, the authors state *"Considerable differences between the two inversions with different a priori emission distributions occurred on the country scale"* but do not explain the different inversions, priors, etc. Supplemental Section 3 provides a good explanation of the inversion framework from Brunner and Henne and I would recommend incorporating some of that text in the manuscript, it would be very useful for the reader.

**2.2 Differences between this work and Miller et al., (2010)?**

The authors note that *"Miller et al., (2010) calculated global emissions of HCFC-23 using the same AGAGE 12-box model as used here, but with a different Bayesian inverse modelling framework."* (Section 3.2). However, the authors do not seem to explain the differences between the inverse modelling frameworks. This seems like a crucial detail because the HFC-23 emissions from Miller et al. are outside the errorbars presented here (Figure 3). This comment seems to go back to my previous comment on the structure of the manuscript. The authors spent a lot of time explaining the measurements but, from my reading, it doesn't seem like the measurements are what give them different emissions.

**2.3 Speculative statements**

There are a number of statements that seem overly speculative and it's not clear that they are supported by the analysis. Here I list two rather provocative statements:

**Statement (Section 3.2):** *"We also note that this minimum occurred during the global financial crisis of 2007-2009 and in fact HFC-23 emissions mirror global GDP growth rates for the years before and after 2009*

*(https://data.worldbank.or/indicator.NY.GDP.MKTP.CD). We can only speculate that this may have reduced the overall demand for PTFE, thereby impacting global HCFC-22 production and the co-produced HFC-23."*

**Statement (Conclusions):** *"With the support of the Chinese Government, 13 new destruction facilities at 15 HCFC-22 production lines not covered by CDM were started in 2014 (UNEP, (2017a). The timing of these new initiatives is consistent with the most recent reduction (2015-2016) of global HFC-23 emissions, although we cannot confirm a direct link."*

---

## Referee Comment (RC2) · Anonymous Referee #2 · 11 Dec 2017

This manuscript reports an update of the evolution of global mole fractions and emissions of HFC-23, an important greenhouse gas. The findings are novel and interesting and the overall work is certainly of sufficient quality for publication in ACP. I would however suggest a) clarifying the reasons for publishing this gas separately from the 2017 Simmonds paper on HFCs and HCFCs and b) the following changes:

L33-36 I'm not sure what the authors want to say here. Does this mean that the HCFC-22 production process has been releasing an increasingly high fraction of HFC-23 or are other sources important, too? This should be clarified in a concise way.

L47-49 So which regions are likely to be responsible for the other 98.5 %?

[Figure]

L53-58 This sentence is very long. Consider splitting it up to improve readability.

L75 The date of that reference is inconsistent with the one in the reference list.

L75-91 This section is a rather abrupt change from the previous and could benefit from short introductory sentence or sub-heading. I would also suggest moving to after the next paragraph.

L88-91 I'm not sure why the extra details are relevant. Surprisingly, the reference to Simmonds et al., 2017, who reported HCFC-22 observations and emissions from the same network (and repeatedly discussed HFC-23), is missing, as are any other papers on HCFC-22.

L126-169 There are quite a lot of technical details in this section, which do not contribute to the main messages of the paper. These methods are also well established and I suggest moving large parts of the section to the supplement to improve readability.

L203 I would be interested to know the methodology used to calculate this "liberal estimate". Is there a reference and could more information specifically on HFC-23 and HCFC-22 be given in the supplement?

L277 The annual growth probably relates to all of 2016, not just the end of that year?

L293-296 The authors should make it clear that they have already published the HCFC-22 data they refer to here.

L314-316 It is not clear whether this agrees with Carpenter and Reimann (2014) or not.

L469 Please correct: un-abated.

L779-781 As published in Simmonds et al., 2017?

L800-816 Missing from the two figures is previously published data and the respective sections of the manuscript would also benefit from a discussion of how mole fractions

compare with published data. The axes on the inset of the first figure are very hard to read and the firn air is missing from the legend.

---

## Author Response (AR1)

**Recent increases in the atmospheric growth rate and emissions of HFC-23 (CHF$_3$) and the link to HCFC-22 (CHClF$_2$) production**

Peter G. Simmonds[1], Matthew Rigby[1], Archie McCulloch[1], Martin K. Vollmer[2], Stephan Henne[2], Jens Mühle[3], Simon O'Doherty[1], Alistair J. Manning[4], Paul B. Krummel[5], Paul J. Fraser[5], Dickon Young[1], Ray F. Weiss[3], Peter K. Salameh[3], Christina M. Harth[3], Stefan Reimann[2], Cathy M. Trudinger[5], L. Paul Steele[5], Ray H. J. Wang[6], Diane J. Ivy[7], Ronald G. Prinn[7], Blagoj Mitrevski[5], and David M. Etheridge[5].

[revised manuscript text omitted]
 emissions (Gg yr$^{-1}$) ±1 sigma (σ) SD. | HFC-23 global mean mole fraction (pmol mol$^{-1}$) ±1 sigma (σ) SD. | HFC-23 global growth rate (pmol mol$^{-1}$ yr$^{-1}$) ±1 sigma (σ) SD. |
|---|---|---|---|
| 1980 | 4.2 ± 0.7 | 3.9 ± 0.1 | 0.33 ± 0.05 |
| 1981 | 4.2 ± 0.8 | 4.3 ± 0.1 | 0.33 ± 0.05 |
| 1982 | 4.4 ± 0.7 | 4.6 ± 0.1 | 0.35 ± 0.07 |
| 1983 | 5.2 ± 0.7 | 5.0 ± 0.1 | 0.41 ± 0.05 |
| 1984 | 5.6 ± 0.7 | 5.4 ± 0.1 | 0.44 ± 0.05 |
| 1985 | 5.8 ± 0.8 | 5.9 ± 0.1 | 0.45 ± 0.05 |
| 1986 | 5.8 ± 0.7 | 6.3 ± 0.1 | 0.45 ± 0.05 |
| 1987 | 5.9 ± 0.7 | 6.8 ± 0.1 | 0.46 ± 0.05 |
| 1988 | 6.8 ± 0.7 | 7.2 ± 0.1 | 0.52 ± 0.05 |
| 1989 | 7.1 ± 0.7 | 7.8 ± 0.2 | 0.55 ± 0.05 |
| 1990 | 7.0 ± 0.7 | 8.3 ± 0.2 | 0.54 ± 0.05 |
| 1991 | 7.0 ± 0.7 | 8.9 ± 0.2 | 0.54 ± 0.05 |
| 1992 | 7.4 ± 0.6 | 9.4 ± 0.2 | 0.57 ± 0.05 |
| 1993 | 7.9 ± 0.6 | 10.0 ± 0.2 | 0.61 ± 0.04 |
| 1994 | 8.3 ± 0.7 | 10.6 ± 0.2 | 0.64 ± 0.04 |
| 1995 | 8.9 ± 0.6 | 11.3 ± 0.2 | 0.69 ± 0.05 |
| 1996 | 9.6 ± 0.6 | 12.0 ± 0.2 | 0.74 ± 0.04 |
| 1997 | 10.1 ± 0.6 | 12.8 ± 0.3 | 0.77 ± 0.04 |
| 1998 | 10.4 ± 0.7 | 13.6 ± 0.3 | 0.79 ± 0.04 |
| 1999 | 10.9 ± 0.7 | 14.4 ± 0.3 | 0.82 ± 0.04 |
| 2000 | 10.4 ± 0.8 | 15.2 ± 0.3 | 0.76 ± 0.05 |
| 2001 | 9.4 ± 0.7 | 15.9 ± 0.3 | 0.68 ± 0.05 |
| 2002 | 9.5 ± 0.7 | 16.6 ± 0.3 | 0.69 ± 0.05 |
| 2003 | 10.3 ± 0.8 | 17.3 ± 0.3 | 0.77 ± 0.05 |
| 2004 | 11.8 ± 0.8 | 18.1 ± 0.3 | 0.90 ± 0.05 |
| 2005 | 13.2 ± 0.8 | 19.1 ± 0.4 | 1.01 ± 0.05 |
| 2006 | 13.3 ± 0.8 | 20.1 ± 0.4 | 0.99 ± 0.05 |
| 2007 | 11.7 ± 0.7 | 21.0 ± 0.4 | 0.85 ± 0.04 |
| 2008 | 11.2 ± 0.6 | 21.9 ± 0.4 | 0.78 ± 0.03 |
| 2009 | 9.6 ± 0.6 | 22.6 ± 0.4 | 0.68 ± 0.03 |
| 2010 | 10.4 ± 0.6 | 23.3 ± 0.4 | 0.74 ± 0.03 |
| 2011 | 11.6 ± 0.6 | 24.1 ± 0.5 | 0.85 ± 0.03 |
| 2012 | 12.9 ± 0.6 | 25.0 ± 0.5 | 0.96 ± 0.03 |
| 2013 | 14.0 ± 0.6 | 26.0 ± 0.5 | 1.04 ± 0.03 |
| 2014 | 14.5 ± 0.6 | 27.0 ± 0.5 | 1.05 ± 0.03 |
| 2015 | 13.1 ± 0.7 | 28.0 ± 0.5 | 0.95 ± 0.03 |
| 2016 | 12.7 ± 0.6 | 28.9 ± 0.6 | 0.94 ± 0.03 |

Note: Data are tabulated as annual mean mid-year values.

Earlier emissions estimates (1930-1979) determined from the AGAGE 12-box model are listed in Supplementary Material (6), Table S5.

Table 3. Annual mean global HCFC-22 emissions, mole fractions, and growth rates, derived from the AGAGE 12-box model.

| Year | HCFC-22 global annual emissions (Gg yr$^{-1}$) ±1 sigma (σ) SD. | HCFC-22 global mean mole fraction (pmol mol$^{-1}$) ±1 sigma (σ) SD. | HCFC-22 global growth rate (pmol mol$^{-1}$ yr$^{-1}$) ±1 sigma (σ) SD. |
|---|---|---|---|
| 1980 | 116.8 ± 17.6 | 41.6 ± 1.0 | 4.1 ± 0.9 |
| 1981 | 123.4 ± 18.8 | 45.8 ± 1.2 | 4.2 ± 0.8 |
| 1982 | 125.7 ± 15.7 | 49.9 ± 1.3 | 4.0 ± 0.8 |
| 1983 | 125.2 ± 20.1 | 53.7 ± 0.9 | 3.6 ± 1.0 |
| 1984 | 136.8 ± 18.1 | 57.3 ± 1.1 | 4.2 ± 0.6 |
| 1985 | 166.1 ± 17.0 | 62.2 ± 1.0 | 5.7 ± 0.7 |
| 1986 | 174.2 ± 19.3 | 68.5 ± 1.0 | 5.5 ± 0.7 |
| 1987 | 155.4 ± 20.2 | 72.9 ± 1.2 | 4.1 ± 0.7 |
| 1988 | 177.8 ± 19.6 | 77.2 ± 1.0 | 5.0 ± 0.7 |
| 1989 | 198.0 ± 19.8 | 82.9 ± 1.0 | 6.0 ± 0.7 |
| 1990 | 209.1 ± 19.7 | 89.1 ± 1.2 | 6.2 ± 0.6 |
| 1991 | 212.2 ± 21.4 | 95.1 ± 1.2 | 5.8 ± 0.7 |
| 1992 | 207.1 ± 23.9 | 100.5 ± 1.3 | 5.0 ± 0.6 |
| 1993 | 214.6 ± 22.8 | 105.4 ± 1.3 | 5.0 ± 0.6 |
| 1994 | 222.7 ± 22.9 | 110.4 ± 1.3 | 5.2 ± 0.5 |
| 1995 | 241.4 ± 26.9 | 115.9 ± 1.3 | 5.7 ± 0.5 |
| 1996 | 230.1 ± 24.3 | 121.4 ± 1.3 | 4.8 ± 0.5 |
| 1997 | 238.3 ± 24.1 | 125.8 ± 1.3 | 5.0 ± 0.5 |
| 1998 | 256.0 ± 27.2 | 131.7 ± 1.3 | 5.7 ± 0.4 |
| 1999 | 251.8 ± 28.4 | 136.8 ± 1.4 | 5.0 ± 0.3 |
| 2000 | 275.1 ± 28.7 | 142.0 ± 1.4 | 5.8 ± 0.2 |
| 2001 | 275.3 ± 28.8 | 147.9 ± 1.5 | 5.5 ± 0.2 |
| 2002 | 280.7 ± 31.6 | 153.1 ± 1.6 | 5.3 ± 0.2 |
| 2003 | 286.3 ± 30.1 | 158.3 ± 1.6 | 5.3 ± 0.2 |
| 2004 | 292.1 ± 30.6 | 163.7 ± 1.7 | 5.3 ± 0.2 |
| 2005 | 312.3 ± 34.6 | 169.2 ± 1.6 | 6.1 ± 0.2 |
| 2006 | 334.3 ± 35.0 | 175.9 ± 1.7 | 7.2 ± 0.2 |
| 2007 | 355.6 ± 35.2 | 183.6 ± 1.8 | 8.0 ± 0.2 |
| 2008 | 372.9 ± 38.4 | 191.9 ± 1.9 | 7.9 ± 0.2 |
| 2009 | 368.6 ± 39.7 | 199.2 ± 2.0 | 7.4 ± 0.2 |
| 2010 | 385.8 ± 41.3 | 206.8 ± 2.0 | 7.4 ± 0.3 |
| 2011 | 373.1 ± 41.3 | 213.7 ± 2.1 | 6.2 ± 0.2 |
| 2012 | 373.2 ± 45.5 | 219.3 ± 2.2 | 5.5 ± 0.2 |
| 2013 | 369.6 ± 44.1 | 224.7 ± 2.3 | 5.0 ± 0.2 |
| 2014 | 373.9 ± 45.7 | 229.5 ± 2.3 | 4.6 ± 0.2 |
| 2015 | 364.2 ± 47.7 | 233.7 ± 2.2 | 3.9 ± 0.2 |
| 2016 | 370.3 ± 45.9 | 237.5 ± 2.2 | 3.9 ± 0.2 |

Note: Data are tabulated as annual mean mid-year values. These HCFC-22 global emissions estimates are updates of the HCFC-22 emissions reported in Simmonds et al. (2017) for the period 1995-2015.

Table 4: European HFC-23 emissions (tonne, Mg) by country/region: $E_a$ *a priori*, $E_b$ *a posteriori* emissions, $f_a$ fraction of *a priori* emissions from factory locations, $f_b$ fraction of *a posteriori* emissions from factory locations. All values represent averages from both inversions using different *a priori* distributions.

| year | Germany | | | | France | | | | Italy | | | |
|------|---------|---------|-----|-----|---------|---------|-----|-----|---------|---------|-----|-----|
| | $E_a$ (Mg/yr) | $E_b$ (Mg/yr) | $f_a$ (%) | $f_b$ (%) | $E_a$ (Mg/yr) | $E_b$ (Mg/yr) | $f_a$ (%) | $f_b$ (%) | $E_a$ (Mg/yr) | $E_b$ (Mg/yr) | $f_a$ (%) | $f_b$ (%) |
| 2009 | 8±16 | 34±12 | 30 | 49 | 15±31 | 2±3.2 | 88 | 6 | 8.4±17 | 34±8.7 | 26 | 52 |
| 2010 | 7.6±15 | 19±14 | 27 | 13 | 12±23 | 15±5.1 | 84 | 86 | 9±18 | 48±9.3 | 26 | 45 |
| 2011 | 8±16 | 44±13 | 33 | 58 | 7.7±15 | 10±3.4 | 70 | 73 | 9.2±18 | 34±11 | 26 | 43 |
| 2012 | 7.6±15 | 32±9.7 | 30 | 40 | 8.1±16 | 8.3±3.4 | 76 | 73 | 9.1±18 | 25±8.6 | 26 | 31 |
| 2013 | 7.2±14 | 16±9.4 | 28 | 58 | 9.2±18 | 27±16 | 82 | 93 | 9.3±19 | 47±24 | 26 | 29 |
| 2014 | 7.1±14 | 20±9.2 | 30 | 27 | 9.4±19 | 10±2.3 | 84 | 85 | 9.5±19 | 32±14 | 26 | 32 |
| 2015 | 6.6±13 | 12±8.7 | 29 | 14 | 9.5±19 | 17±3.9 | 86 | 92 | 9.8±20 | 37±19 | 26 | 24 |
| 2016 | 6.6±13 | 19±9.1 | 29 | 26 | 9.5±19 | 9.9±3.4 | 86 | 85 | 9.8±20 | 23±10 | 26 | 39 |

| year | Benelux | | | | United Kingdom | | | | Iberian Peninsula | | | |
|------|---------|---------|-----|-----|---------|---------|-----|-----|---------|---------|-----|-----|
| | $E_a$ (Mg/yr) | $E_b$ (Mg/yr) | $f_a$ (%) | $f_b$ (%) | $E_a$ (Mg/yr) | $E_b$ (Mg/yr) | $f_a$ (%) | $f_b$ (%) | $E_a$ (Mg/yr) | $E_b$ (Mg/yr) | $f_a$ (%) | $f_b$ (%) |
| 2009 | 13±27 | 25±8.8 | 98 | 99 | 3.8±7.6 | 5.3±3.1 | 84 | 87 | 97±190 | 56±29 | 57 | 10 |
| 2010 | 34±67 | 16±7.1 | 99 | 98 | 1.2±2.3 | 2.8±1.9 | 48 | 71 | 130±260 | 120±33 | 66 | 52 |
| 2011 | 15±29 | 21±16 | 97 | 97 | 1±2.1 | 1.5±1.7 | 39 | 21 | 83±170 | 75±27 | 50 | 27 |
| 2012 | 11±22 | 53±13 | 96 | 99 | 0.92±1.8 | 2±1.6 | 26 | 46 | 74±150 | 46±27 | 45 | 43 |
| 2013 | 16±33 | 94±13 | 98 | 100 | 1.1±2.1 | 2.1±1.8 | 29 | 48 | 64±130 | 110±27 | 38 | 22 |
| 2014 | 3.8±7.6 | 11±6.3 | 85 | 94 | 1.3±2.5 | 6.8±2.4 | 35 | 73 | 59±120 | 55±27 | 35 | 45 |
| 2015 | 8.7±17 | 20±12 | 94 | 97 | 1.4±2.7 | 2.5±2.1 | 34 | 37 | 48±96 | 19±18 | 26 | 9 |
| 2016 | 8.7±17 | 45±11 | 94 | 99 | 1.4±2.7 | 3.8±2.2 | 34 | 42 | 48±96 | 69±24 | 26 | 33 |

Note : **Benelux** (Belgium, the Netherlands, and Luxembourg).

[Figure]

Figure 1. HFC-23 modelled mole fractions for the four equal-mass latitudinal subdivisions of the global atmosphere calculated from the 12-box model and the *in situ* records from the AGAGE core sites (points with error bars), firn air (squares, red SH, blue NH), old NH air data (blue circles, only shown for times without NH *in situ* data), and CGAA data (red circles, only shown for times without SH in situ data). Lower box shows the annual growth rates (global – blue solid line with uncertainty band - and individual semi-hemispheres – dashed lines) in pmol mol$^{-1}$ yr$^{-1}$. Figure 1 inset, compares the annual mean mole fractions of HFC-23 and HCFC-22 recorded at Mace Head and Cape Grim from 2007-2016.

[Figure]

Figure 2. HCFC-22 modelled mole fractions for the four equal-mass latitudinal subdivisions of
the global atmosphere calculated from the 12-box model and the *in situ* records from the
AGAGE core sites (points with error bars) and CGAA data (red circles, only shown for times
without *in situ* data) and old NH air data (blue circles, only shown for times without NH *in situ*
data). Lower box shows the annual growth rates (global – blue solid line with uncertainty band -
and individual semi-hemispheres – dashed lines) in pmol mol$^{-1}$ yr$^{-1}$.

[Figure]

Figure 3. Global emissions of HFC-23 calculated from the AGAGE 12-box model (blue line and
shading, 1 σ uncertainty). Data are plotted as annual mean values, centred on the middle of each
year. The purple line shows bottom-up HFC-23 estimates of emissions from Miller et al. (2010),
the yellow line HFC-23 emissions estimates from Montzka et al. (2010) and the grey line from
Carpenter and Reimann (WMO 2014). The green line shows emissions reports by Annex-1
countries. The red line shows the bottom-up estimated global emissions developed here and
discussed in (Supplementary Material 3).

[Figure]

Figure 4. Global emissions of HCFC-22 calculated from the AGAGE 12-box model (blue line
and shading, 1σ uncertainty). Data are plotted as annual mean mid-year values. The green line
shows bottom-up HCFC-22 estimates of emissions from Miller et al. (2010). Red and purple
lines show HCFC-22 emissions calculated from AGAGE and NOAA data/models respectively,
reported in WMO 2014 (Carpenter and Reimann, 2014).

[Figure]

Figure 5a. HFC-23 global and Chinese emissions estimates and the percentage of Chinese emissions contributing to the global total (dashed line). Compiled from Table S4 in Supplementary Material (3).

[Figure]

Figure 5b. HCFC-22 emissions estimates of global and Chinese production and the percentage of Chinese production as a fraction of the global total (dashed line). Compiled from Table S3 in Supplementary Material (3).

[Figure]

Figure 6: Temporal evolution of national/regional emissions of HFC-23: solid bars and error bars give *a posteriori* emissions using the two sets of *a priori* emissions (grey lines). The two approaches (green and orange bars) spatially distribute these different bottom-up estimates (see section 2.7.3). Blue horizontal lines give the estimates of Keller et al. (2011) for their Bayesian (light blue) and point source (dark blue) estimate; a) Germany (DE), b) Italy (IT), c) France (FR), d) Spain and Portugal (ES+PR), e) United Kingdom (UK), f) Benelux countries (Netherlands, Belgium, Luxembourg, NE+BE+LX).

**Recent increases in the growth rate and emissions of HFC-23 (CHF$_3$) and the link to HCFC-22 (CHClF$_2$) production**

Peter G. Simmonds[1], Matthew Rigby[1], Archie McCulloch[1], Martin K. Vollmer[2], Stephan Henne[2], Jens Mühle[3], Simon O'Doherty[1], Alistair J. Manning[4], Paul B. Krummel[5], Paul J. Fraser[5], Dickon Young[1], Ray F. Weiss[3], Peter K. Salameh[3], Christina M. Harth[3], Stefan Reimann[2], Cathy M. Trudinger[5], L. Paul Steele[5], Ray H. J. Wang[6], Diane J. Ivy[7], Ronald G. Prinn[7], Blagoj Mitrevski[5], and David M. Etheridge[5].

[1] Atmospheric Chemistry Research Group, University of Bristol, Bristol, UK
[2] Swiss Federal Laboratories for Materials Science and Technology, Laboratory for Air Pollution and Environmental Technology (Empa), Dübendorf, Switzerland,
[3] Scripps Institution of Oceanography (SIO), University of California at San Diego, La Jolla, California, USA
[4] Met Office Hadley Centre, Exeter, UK
[5] Climate Science Centre, Commonwealth Scientific and Industrial Research Organisation (CSIRO) Oceans and Atmosphere, Aspendale, Victoria, Australia
[6] School of Earth, and Atmospheric Sciences, Georgia Institute of Technology, Atlanta, Georgia, USA
[7] Center for Global Change Science, Massachusetts Institute of Technology, Cambridge, Massachusetts, USA

Correspondence to: P.G. Simmonds (petergsimmonds@aol.com)

**Supplementary Material (1): AGAGE Instrumentation and Measurement Techniques**

Typically for each measurement, the analytes from two litres of air are collected on the microtrap and, after fractionated distillation, purification and transfer, are desorbed onto a single main capillary chromatography column (CP-PoraBOND Q, 0.32 mm ID × 25 m, 5 μm, Agilent Varian Chrompack, batch-made for AGAGE applications) purged with helium (research grade 6.0) that is further purified using a heated helium purifier (HP2, VICI, USA). Separation and detection of the compounds are achieved by using Agilent Technology GCs (model 6890N) and quadrupole mass spectrometers in selected ion mode (initially model 5973 series, progressively converted to 5975C over the later years).

The quaternary standards are whole-air samples, pressurized into 34 L internally electropolished stainless steel canisters (Essex Industries, USA). They are filled by the responsible station scientist and/or on-site station personnel who are in charge of the respective AGAGE remote sites using modified oil-free diving compressors (SA-3 and SA-6, RIX Industries, USA) to ~60 bar (older canisters to ~40 bar). Cape Grim is an exception, where the canisters used for quaternary standard purposes are filled cryogenically. This method of cryogenically collecting large volumes of ambient air is the same as that is used for collecting air for the CGAA and measurements of many atmospheric trace species in air samples collected in this manner show that the trace gas composition of the air is well preserved (Fraser et al., 1991, 2016; Langenfelds et al., 1996, 2003). The on-site quaternary standards are compared weekly to tertiary standards from the central calibration facility at the Scripps Institution of Oceanography (SIO) in order to propagate the primary calibration scales and assess any long-term drifts. These tertiary standards are filled with ambient air in Essex canisters under "baseline" clean air conditions at Trinidad Head or at La Jolla (California) and are measured at SIO against secondary ambient air standards (to obtain an "out" value) before they are shipped to individual AGAGE sites. We define "baseline" as air masses that are representative of the unpolluted marine boundary layer, uninfluenced by recent local or regional emissions. After their on-site deployment they are again measured at SIO to obtain an "in" value, to assess any possible drifts. They are also measured on-site against the previous and next tertiaries. The secondary standards and the synthetic primary standards at SIO provide the core of the AGAGE calibration system (Prinn et al., 2000; Miller et al., 2008).

The GC-MS-Medusa measurement precisions for HFC-23 and HCFC-22 are determined as the precisions of replicate measurements of the quaternary standards over twice the time interval as for sample-standard comparisons (Miller et al., 2008). Accordingly, they are upper-limit estimates of the precisions of the sample-standard comparisons. Typical daily precisions for each compound vary with abundance and individual instrument performance over time. Average percentage relative standard deviation (% RSD) between 2007 and 2016 were: HFC-23 (0.1%-1.9%, average 0.7%); and for HCFC-22 (0.1%-2.5%, average 0.6%).

**Supplementary Material (2): Firn Air Depth Profiles, Analyses of the CGAA and old Northern Hemisphere (NH) air samples**

In this section we illustrate in Figures S1 the depth profiles for HFC-23 in the polar firn and in Figure S2 we show three independent analyses of the data from the CGAA. Tables S1 and S2 also list the actual data used to construct these figures.

Figure S1. Depth profiles for HFC-23 in polar firn. DSSW20K and SPO-01 are Antarctic sites and NEEM-08 is from Greenland. The modelled mole fractions correspond to the optimized emissions history using an inversion and firn air model developed at CSIRO.

[Figure]

Figure S2. Comparison of three analysis sets of HFC-23 in the Cape Grim Air Archive

[Figure]

A series of old Northern Hemisphere (NH) air samples were mostly collected during clean air conditions but not with the purpose of creating a consistent air archive. Therefore, a stepwise tightening filtering algorithm was applied to the measurement results based on their deviations from a fit through all data (including in situ data). Due to the scarcity of the Northern Hemisphere HFC-23 data, the filtering of these samples used the fit through the filtered Southern Hemisphere samples as additional guide (with an appropriate time lag related to hemispheric transport). The remaining final NH HFC-23 data showed good agreement with concurrent in situ measurements. (Mühle et al., 2010; Vollmer et al., 2016)

Table S1

Firn air measurement and model results for HFC-23

Abbreviations: m: measured; mf: mole fraction; p: precision (measurement repeatability, 1 sigma); mod: firn air model output with uncertainties

Primary calibration scale for HFC-23: SIO-07

depth: depth in firn air hole from which sample was drawn

Sample Volume: volume of sample used in one analysis on the Medusa-GCMS

Flags used for the decisions on presence of the compound in the sample

Flag 1: Peak size large enough a non-zero positive mole fraction was calculated and reported.

Flag 2: Clear sign of a peak but very small. Mole fraction was calculated by GCWerks either using generally set parameters or using GCWerks special integration

Flag 3: Maybe a peak some baseline disturbance that point to a non-zero signal. In most cases a mole fraction assigned

Flag 4: no sign of a peak at all no change in baseline. Mole fraction definitely smaller than the estimated detection limit for that sample

| Site | Tank_ID | UAN | Parent_UAN | depth | sample volume | m-mf | m-p | n | Flag | mod-mf | mod-min | mod-max | mean | effect |
|---|---|---|---|---|---|---|---|---|---|---|---|---|---|---|
| | | | | [m] | [L] | [ppt] | [ppt] | | | [ppt] | [ppt] | [ppt] | Age | Age |
| DSSW20K | S22L-002 | UAN980141 | UAN980141 | 15.8 | 3 | 11.295 | 0.071 | 3 | 1 | 11.335 | 11.282 | 11.449 | 1995.95 | 1996.26 |
| DSSW20K | MC-05 | UAN980780 | UAN980142 | 29 | 2 | 9.924 | 0.098 | 2 | 1 | 9.968 | 9.856 | 10.1 | 1993.63 | 1994.04 |
| DSSW20K | CA01674 | UAN980143 | UAN980143 | 37.8 | 3 | 8.796 | 0.041 | 4 | 1 | 8.431 | 8.327 | 8.624 | 1990.64 | 1991.23 |
| DSSW20K | MC-08 | UAN980783 | UAN980144 | 41.7 | 2 | 7.074 | 0.06 | 2 | 1 | 7.075 | 6.954 | 7.307 | 1987.71 | 1988.85 |

| | | | | | | | | | | | | | | |
|---|---|---|---|---|---|---|---|---|---|---|---|---|---|---|
| DSSW20K | MC-09 | UAN980784 | UAN980145 | 44.5 | 2 | 4.49 | 0.018 | 2 | 1 | 4.425 | 4.221 | 4.54 | 1980.51 | 1982.51 |
| DSSW20K | MC-06 | UAN980781 | UAN980146 | 47 | 2 | 1.215 | 0.021 | 2 | 1 | 1.323 | 1.195 | 1.411 | 1966.83 | 1970.73 |
| DSSW20K | MC-04 | UAN980779 | UAN980147 | 49.5 | 2 | 0.196 | 0.033 | 2 | 1 | 0.478 | 0.414 | 0.53 | 1952.95 | 1953.44 |
| DSSW20K | S22L-010 | UAN980148 | UAN980148 | 49.5 | 1.5 | 0.468 | 0.061 | 2 | 1 | 0.478 | 0.414 | 0.53 | 1952.95 | 1953.44 |
| DSSW20K | MC-01 | UAN980776 | UAN980150 | 52 | 2 | 0.128 | 0.011 | 2 | 1 | 0.478 | 0.414 | 0.53 | 1952.95 | 1953.44 |
| DSSW20K | MC-10 | UAN980785 | UAN980149 | 52 | 1 | 0.451 | 0.064 | 2 | 1 | 0.478 | 0.414 | 0.53 | 1952.95 | 1953.44 |
| DSSW20K | S22L-007 | UAN980151 | UAN980151 | 52 | 2 | 0.115 | 0 | 1 | 1 | 0.137 | 0.137 | 0.292 | 1938.93 | 1937.62 |
| NEEM-2008 | S300-B15 | UAN999698 | UAN999698 | 0 | 1.5 | 22.462 | 0.104 | 2 | 1 | 0.137 | 0.137 | 0.292 | 1938.93 | 1937.62 |
| NEEM-2008 | S300-B13 | UAN999697 | UAN999697 | 20 | 1.5 | 21.502 | 0.189 | 2 | 1 | 0.137 | 0.137 | 0.292 | 1938.93 | 1937.62 |
| NEEM-2008 | S300-B11 | UAN999695 | UAN999695 | 50 | 1.5 | 18.067 | 0.067 | 2 | 1 | 0.137 | 0.137 | 0.292 | 1938.93 | 1937.62 |
| NEEM-2008 | S300-B12 | UAN999696 | UAN999696 | 64 | 1.5 | 12.602 | 0.1 | 2 | 1 | 0.137 | 0.137 | 0.292 | 1938.93 | 1937.62 |
| NEEM-2008 | S300-B16 | UAN999699 | UAN999699 | 68 | 1.5 | 4.322 | 0.012 | 2 | 1 | 0.137 | 0.137 | 0.292 | 1938.93 | 1937.62 |
| NEEM-2008 | S300-B18 | UAN999701 | UAN999701 | 70 | 1.5 | 2.204 | 0.051 | 2 | 1 | 0.137 | 0.137 | 0.292 | 1938.93 | 1937.62 |
| NEEM-2008 | S300-B19 | UAN999702 | UAN999702 | 72 | 1.5 | 1.196 | 0.047 | 2 | 1 | 0.137 | 0.137 | 0.292 | 1938.93 | 1937.62 |
| NEEM-2008 | S300-B17 | UAN999700 | UAN999700 | 74 | 1.5 | 0.575 | 0.05 | 2 | 1 | 0.137 | 0.137 | 0.292 | 1938.93 | 1937.62 |
| NEEM-2008 | S300-B20 | UAN999703 | UAN999703 | 76 | 1.5 | 0.541 | 0.011 | 2 | 1 | NaN | NaN | NaN | NaN | NaN |
| SPO | S300-A23 | UAN996580 | UAN993582 | 119.87 | 1 | 0.994 | 0.04 | 2 | 1 | 21.391 | 21.182 | 21.537 | 2006.96 | 2007 |

Table S1

Firn air measurement and model results for HFC-23

Abbreviations: m: measured; mf: mole fraction; p: precision (measurement repeatability, 1 sigma); mod: firn air model output with uncertainties

Primary calibration scale for HFC-23: SIO-07

depth: depth in firn air hole from which sample was drawn

Sample Volume: volume of sample used in one analysis on the Medusa-GCMS

Flags used for the decisions on presence of the compound in the sample

Flag 1: Peak size large enough a non-zero positive mole fraction was calculated and reported.

Flag 2: Clear sign of a peak but very small. Mole fraction was calculated by GCWerks either using generally set parameters or using GCWerks special integration

Flag 3: Maybe a peak some baseline disturbance that point to a non-zero signal. In most cases a mole fraction assigned

Flag 4: no sign of a peak at all no change in baseline. Mole fraction definitely smaller than the estimated detection limit for that sample

| Site | Tank_ID | UAN | Parent_UAN | sample depth | volume | m-mf | m-p | n | Flag | HFC-23 mod-mf | mod-min | mod-max | mean | effect |
|---|---|---|---|---|---|---|---|---|---|---|---|---|---|---|
| | | | | [m] | [L] | [ppt] | [ppt] | | | [ppt] | [ppt] | [ppt] | Age | Age |
| DSSW20K | S22L-002 | UAN980141 | UAN980141 | 15.8 | 3 | 11.295 | 0.071 | 3 | 1 | 11.335 | 11.282 | 11.449 | 1995.95 | 1996.26 |
| DSSW20K | MC-05 | UAN980780 | UAN980142 | 29 | 2 | 9.924 | 0.098 | 2 | 1 | 9.968 | 9.856 | 10.1 | 1993.63 | 1994.04 |
| DSSW20K | CA01674 | UAN980143 | UAN980143 | 37.8 | 3 | 8.796 | 0.041 | 4 | 1 | 8.431 | 8.327 | 8.624 | 1990.64 | 1991.23 |
| DSSW20K | MC-08 | UAN980783 | UAN980144 | 41.7 | 2 | 7.074 | 0.06 | 2 | 1 | 7.075 | 6.954 | 7.307 | 1987.71 | 1988.85 |

| | | | | | | | | | | | | | | |
|---|---|---|---|---|---|---|---|---|---|---|---|---|---|---|
| DSSW20K | MC-09 | UAN980784 | UAN980145 | 44.5 | 2 | 4.49 | 0.018 | 2 | 1 | 4.425 | 4.221 | 4.54 | 1980.51 | 1982.51 |
| DSSW20K | MC-06 | UAN980781 | UAN980146 | 47 | 2 | 1.215 | 0.021 | 2 | 1 | 1.323 | 1.195 | 1.411 | 1966.83 | 1970.73 |
| DSSW20K | MC-04 | UAN980779 | UAN980147 | 49.5 | 2 | 0.196 | 0.033 | 2 | 1 | 0.478 | 0.414 | 0.53 | 1952.95 | 1953.44 |
| DSSW20K | S22L-010 | UAN980148 | UAN980148 | 49.5 | 1.5 | 0.468 | 0.061 | 2 | 1 | 0.478 | 0.414 | 0.53 | 1952.95 | 1953.44 |
| DSSW20K | MC-01 | UAN980776 | UAN980150 | 52 | 2 | 0.128 | 0.011 | 2 | 1 | 0.478 | 0.414 | 0.53 | 1952.95 | 1953.44 |
| DSSW20K | MC-10 | UAN980785 | UAN980149 | 52 | 1 | 0.451 | 0.064 | 2 | 1 | 0.478 | 0.414 | 0.53 | 1952.95 | 1953.44 |
| DSSW20K | S22L-007 | UAN980151 | UAN980151 | 52 | 2 | 0.115 | 0 | 1 | 1 | 0.137 | 0.137 | 0.292 | 1938.93 | 1937.62 |
| NEEM-2008 | S300-B15 | UAN999698 | UAN999698 | 0 | 1.5 | 22.462 | 0.104 | 2 | 1 | 0.137 | 0.137 | 0.292 | 1938.93 | 1937.62 |
| NEEM-2008 | S300-B13 | UAN999697 | UAN999697 | 20 | 1.5 | 21.502 | 0.189 | 2 | 1 | 0.137 | 0.137 | 0.292 | 1938.93 | 1937.62 |
| NEEM-2008 | S300-B11 | UAN999695 | UAN999695 | 50 | 1.5 | 18.067 | 0.067 | 2 | 1 | 0.137 | 0.137 | 0.292 | 1938.93 | 1937.62 |
| NEEM-2008 | S300-B12 | UAN999696 | UAN999696 | 64 | 1.5 | 12.602 | 0.1 | 2 | 1 | 0.137 | 0.137 | 0.292 | 1938.93 | 1937.62 |
| NEEM-2008 | S300-B16 | UAN999699 | UAN999699 | 68 | 1.5 | 4.322 | 0.012 | 2 | 1 | 0.137 | 0.137 | 0.292 | 1938.93 | 1937.62 |
| NEEM-2008 | S300-B18 | UAN999701 | UAN999701 | 70 | 1.5 | 2.204 | 0.051 | 2 | 1 | 0.137 | 0.137 | 0.292 | 1938.93 | 1937.62 |
| NEEM-2008 | S300-B19 | UAN999702 | UAN999702 | 72 | 1.5 | 1.196 | 0.047 | 2 | 1 | 0.137 | 0.137 | 0.292 | 1938.93 | 1937.62 |
| NEEM-2008 | S300-B17 | UAN999700 | UAN999700 | 74 | 1.5 | 0.575 | 0.05 | 2 | 1 | 0.137 | 0.137 | 0.292 | 1938.93 | 1937.62 |
| NEEM-2008 | S300-B20 | UAN999703 | UAN999703 | 76 | 1.5 | 0.541 | 0.011 | 2 | 1 | NaN | NaN | NaN | NaN | NaN |
| SPO | S300-A23 | UAN996580 | UAN993582 | 119.87 | 1 | 0.994 | 0.04 | 2 | 1 | 21.391 | 21.182 | 21.537 | 2006.96 | 2007 |

Table S2

Cape Grim Air Archive (CGAA) Results for HFC-23 from three analysis periods

Results are reported as dry air mole fractions for abundance (c) and precisions (p)

Measurements are conducted on the CSIRO Aspendale-9 GCMS-Medusa

Primary calibration scales: CFC-23: SIO-07

Notes for 2006: Measurements by B. R. Miller, L. Porter, L. P. Steele, P. B. Krummel. Results by peak area. Standard is G-141 with assigned values: CFC-23: 19.648 ppt

Notes for 2011: Measurements by D. Ivy, L. P. Steele, P. B. Krummel, M. Leist, Results by peak area. Standard is G-181 with assigned values: CFC-23: 23.1456 ppt

Notes for 2016: Measurements by M. K. Vollmer, L. P. Steele, B. Mitrevski, P. B. Krummel, Results by peak area. Standard is E-146S with assigned values: CFC-23: 31.4808 ppt

| sampleID | time | year | month | day | c_mean | p_mean | c_2006 | p_2006 | n_2006 | c_2011 | p_2011 | n_2011 | c_2016 | p_2016 | n_2016 |
| --- | --- | --- | --- | --- | --- | --- | --- | --- | --- | --- | --- | --- | --- | --- | --- |
| | fractional | | | | [ppt] | [ppt] | [ppt] | [ppt] | | [ppt] | [ppt] | | [ppt] | [ppt] | |
| UAN780001 | 1978.315 | 1978 | 4 | 26 | 2.992 | 0.081 | 3.065 | 0.114 | 3 | 2.918 | 0.048 | 3 | NaN | NaN | NaN |
| UAN780002 | 1978.512 | 1978 | 7 | 7 | 3.116 | 0.018 | 3.178 | 0.023 | 3 | 3.037 | 0.009 | 3 | 3.133 | 0.023 | 3 |
| UAN790001 | 1979.099 | 1979 | 2 | 6 | 3.282 | 0.025 | 3.363 | 0.027 | 3 | 3.223 | 0.04 | 3 | 3.259 | 0.008 | 3 |
| UAN910377 | 1984.053 | 1984 | 1 | 20 | 5.453 | 0.044 | 5.453 | 0.044 | 2 | NaN | NaN | NaN | NaN | NaN | NaN |
| UAN840004 | 1984.391 | 1984 | 5 | 23 | 5.226 | 0.023 | 5.2 | 0.034 | 4 | 5.066 | 0.021 | 4 | 5.411 | 0.014 | 3 |
| UAN860001 | 1986.099 | 1986 | 2 | 6 | 5.746 | 0.048 | 5.835 | 0.046 | 4 | 5.71 | 0.086 | 4 | 5.694 | 0.012 | 3 |
| UAN860005 | 1986.863 | 1986 | 11 | 12 | 6.275 | 0.142 | 6.275 | 0.142 | 4 | NaN | NaN | NaN | NaN | NaN | 3 |

| | | | | | | | | | | | | | | | |
|---|---|---|---|---|---|---|---|---|---|---|---|---|---|---|---|
| UAN870006 | 1987.403 | 1987 | 5 | 28 | 6.456 | 0.06 | 6.493 | 0.039 | 4 | 6.418 | 0.08 | 6 | NaN | NaN | NaN |
| UAN880003 | 1988.47 | 1988 | 6 | 21 | 6.651 | 0.058 | 6.646 | 0.065 | 6 | 6.657 | 0.051 | 3 | NaN | NaN | NaN |
| UAN880002 | 1988.47 | 1988 | 6 | 21 | 6.834 | 0.037 | 6.834 | 0.037 | 4 | NaN | NaN | NaN | NaN | NaN | 3 |
| UAN890002 | 1989.299 | 1989 | 4 | 20 | 7.466 | 0.04 | 7.466 | 0.04 | 4 | NaN | NaN | NaN | NaN | NaN | NaN |
| UAN890004 | 1989.378 | 1989 | 5 | 19 | 7.24 | 0.083 | NaN | NaN | NaN | 7.24 | 0.083 | 8 | NaN | NaN | 3 |
| UAN890005 | 1989.852 | 1989 | 11 | 8 | 7.589 | 0.064 | 7.589 | 0.064 | 4 | NaN | NaN | NaN | NaN | NaN | NaN |
| UAN900027 | 1990.127 | 1990 | 2 | 16 | 7.786 | 0.145 | 7.786 | 0.145 | 4 | NaN | NaN | NaN | NaN | NaN | NaN |
| UAN900048 | 1990.315 | 1990 | 4 | 26 | 7.932 | 0.112 | 7.969 | 0.06 | 5 | 7.894 | 0.164 | 3 | NaN | NaN | 3 |
| UAN910361 | 1991.658 | 1991 | 8 | 29 | 8.695 | 0.06 | 8.775 | 0.057 | 3 | 8.592 | 0.09 | 3 | 8.718 | 0.032 | 3 |
| UAN920469 | 1992.211 | 1992 | 3 | 18 | 9.012 | 0.025 | 9.012 | 0.025 | 3 | NaN | NaN | NaN | NaN | NaN | NaN |
| UAN920655 | 1992.727 | 1992 | 9 | 23 | 9.152 | 0.049 | 9.18 | 0.042 | 4 | 9.135 | 0.059 | 5 | 9.14 | 0.046 | 3 |
| UAN930279 | 1993.164 | 1993 | 3 | 2 | 9.321 | 0.023 | 9.353 | 0.024 | 3 | NaN | NaN | NaN | 9.289 | 0.023 | 3 |
| UAN940378 | 1994.112 | 1994 | 2 | 11 | 10.205 | 0.054 | 10.291 | 0.077 | 3 | 10.17 | 0.072 | 3 | 10.154 | 0.012 | 3 |
| UAN940679 | 1994.318 | 1994 | 4 | 27 | 10.06 | 0.043 | 10.06 | 0.043 | 3 | NaN | NaN | NaN | NaN | NaN | NaN |
| UAN941096 | 1994.759 | 1994 | 10 | 4 | 10.259 | 0.192 | 10.259 | 0.192 | 4 | NaN | NaN | NaN | NaN | NaN | NaN |
| UAN950527 | 1995.195 | 1995 | 3 | 13 | 10.596 | 0.023 | 10.596 | 0.023 | 4 | NaN | NaN | NaN | NaN | NaN | NaN |
| UAN950789 | 1995.447 | 1995 | 6 | 13 | 10.788 | 0.059 | 10.82 | 0.088 | 6 | NaN | NaN | NaN | 10.756 | 0.03 | 3 |
| UAN950894 | 1995.584 | 1995 | 8 | 2 | 11.202 | 0.157 | 11.202 | 0.157 | 4 | NaN | NaN | NaN | NaN | NaN | NaN |
| UAN960115 | 1995.811 | 1995 | 10 | 24 | 11.027 | 0.109 | 11.027 | 0.109 | 4 | NaN | NaN | NaN | NaN | NaN | NaN |
| UAN960051 | 1995.923 | 1995 | 12 | 4 | 11.081 | 0.052 | 11.108 | 0.06 | 3 | 11.054 | 0.044 | 5 | NaN | NaN | NaN |
| UAN960957 | 1996.404 | 1996 | 5 | 28 | 11.39 | 0.056 | 11.407 | 0.068 | 5 | NaN | NaN | NaN | 11.373 | 0.043 | 3 |

| UAN961164 | 1996.637 | 1996 | 8 | 21 | 11.705 | 0.123 | 11.705 | 0.123 | 3 | NaN | NaN | NaN | NaN | NaN | NaN |
| UAN961409 | 1996.754 | 1996 | 10 | 3 | 11.668 | 0.04 | 11.695 | 0.074 | 3 | NaN | NaN | NaN | 11.64 | 0.006 | 3 |
| UAN970092 | 1996.885 | 1996 | 11 | 20 | 11.728 | 0.044 | 11.766 | 0.069 | 4 | 11.726 | 0.031 | 4 | 11.693 | 0.033 | 3 |
| UAN970008 | 1997.016 | 1997 | 1 | 7 | 11.809 | 0.06 | 11.871 | 0.114 | 5 | 11.767 | 0.051 | 5 | 11.79 | 0.016 | 3 |
| UAN970011 | 1997.016 | 1997 | 1 | 7 | 11.813 | 0.081 | 11.824 | 0.123 | 3 | NaN | NaN | NaN | 11.802 | 0.039 | 3 |
| UAN970010 | 1997.017 | 1997 | 1 | 7 | 11.812 | 0.062 | 11.812 | 0.062 | 3 | NaN | NaN | NaN | NaN | NaN | NaN |
| UAN970380 | 1997.195 | 1997 | 3 | 13 | 11.934 | 0.063 | 11.97 | 0.085 | 13 | NaN | NaN | NaN | 11.898 | 0.041 | 6 |
| UAN970754 | 1997.255 | 1997 | 4 | 4 | 12.245 | 0.071 | 12.245 | 0.071 | 4 | NaN | NaN | NaN | NaN | NaN | NaN |
| UAN970756 | 1997.408 | 1997 | 5 | 30 | 12.194 | 0.046 | 12.3 | 0.093 | 5 | 12.17 | 0.019 | 4 | 12.112 | 0.027 | 3 |
| UAN971115 | 1997.534 | 1997 | 7 | 15 | 12.263 | 0.057 | 12.281 | 0.044 | 3 | 12.244 | 0.07 | 7 | NaN | NaN | NaN |
| UAN980724 | 1998.285 | 1998 | 4 | 15 | 12.902 | 0.116 | 12.902 | 0.116 | 5 | NaN | NaN | NaN | NaN | NaN | NaN |
| UAN980918 | 1998.479 | 1998 | 6 | 25 | 13.052 | 0.081 | 13.051 | 0.086 | 6 | 13.053 | 0.077 | 5 | NaN | NaN | NaN |
| UAN981563 | 1998.786 | 1998 | 10 | 15 | 13.25 | 0.062 | 13.323 | 0.091 | 15 | NaN | NaN | NaN | 13.177 | 0.032 | 3 |
| UAN991060 | 1999.129 | 1999 | 2 | 17 | 13.606 | 0.089 | 13.606 | 0.089 | 6 | NaN | NaN | NaN | NaN | NaN | NaN |
| UAN991062 | 1999.279 | 1999 | 4 | 13 | 13.717 | 0.067 | 13.737 | 0.026 | 4 | 13.696 | 0.108 | 5 | NaN | NaN | NaN |
| UAN991381 | 1999.59 | 1999 | 8 | 4 | 13.904 | 0.088 | 13.904 | 0.088 | 3 | NaN | NaN | NaN | NaN | NaN | NaN |
| UAN992045 | 1999.874 | 1999 | 11 | 16 | 14.152 | 0.11 | 14.152 | 0.11 | 5 | NaN | NaN | NaN | NaN | NaN | NaN |
| UAN20101335 | 2000.164 | 2000 | 3 | 1 | 14.375 | 0.088 | NaN | NaN | NaN | 14.375 | 0.088 | 6 | NaN | NaN | NaN |
| UAN992982 | 2000.199 | 2000 | 3 | 14 | 14.391 | 0.092 | 14.393 | 0.139 | 6 | 14.426 | 0.111 | 5 | 14.353 | 0.028 | 3 |
| UAN993562 | 2000.744 | 2000 | 9 | 29 | 14.818 | 0.044 | 14.818 | 0.044 | 4 | NaN | NaN | NaN | NaN | NaN | NaN |
| UAN993563 | 2001.038 | 2001 | 1 | 15 | 14.979 | 0.089 | 15.002 | 0.111 | 7 | 14.977 | 0.133 | 4 | 14.957 | 0.022 | 3 |

| | | | | | | | | | | | | | | | | |
|---|---|---|---|---|---|---|---|---|---|---|---|---|---|---|---|---|
| UAN994885 | 2001.545 | 2001 | 7 | 19 | 15.534 | 0.104 | 15.53 | 0.116 | 3 | 15.539 | 0.093 | 5 | NaN | NaN | NaN |
| UAN994886 | 2002.466 | 2002 | 6 | 20 | 16.11 | 0.075 | 16.08 | 0.105 | 4 | 16.13 | 0.086 | 6 | 16.121 | 0.035 | 3 |
| UAN995445 | 2003.129 | 2003 | 2 | 17 | 16.54 | 0.071 | 16.596 | 0.08 | 6 | 16.481 | 0.079 | 5 | 16.544 | 0.053 | 3 |
| UAN996454 | 2003.384 | 2003 | 5 | 21 | 16.802 | 0.071 | 16.802 | 0.071 | 3 | NaN | NaN | NaN | NaN | NaN | NaN |
| UAN996455 | 2003.753 | 2003 | 10 | 3 | 17.018 | 0.062 | 17.081 | 0.096 | 4 | 16.967 | 0.057 | 5 | 17.005 | 0.032 | 3 |
| UAN996456 | 2004.053 | 2004 | 1 | 20 | 17.265 | 0.113 | 17.265 | 0.113 | 3 | NaN | NaN | NaN | NaN | NaN | NaN |
| UAN998318 | 2004.057 | 2004 | 1 | 22 | 17.237 | 0.058 | 17.275 | 0.072 | 5 | 17.24 | 0.059 | 6 | 17.195 | 0.042 | 3 |
| UAN996457 | 2004.268 | 2004 | 4 | 8 | 17.523 | 0.205 | 17.523 | 0.205 | 3 | NaN | NaN | NaN | NaN | NaN | NaN |
| UAN996458 | 2004.459 | 2004 | 6 | 17 | 17.529 | 0.025 | 17.53 | 0.026 | 3 | NaN | NaN | NaN | 17.529 | 0.025 | 3 |
| UAN997089 | 2004.915 | 2004 | 12 | 1 | 17.879 | 0.075 | 17.932 | 0.075 | 16 | 17.826 | 0.075 | 18 | NaN | NaN | NaN |
| UAN997090 | 2005.11 | 2005 | 2 | 10 | 18.009 | 0.037 | 17.991 | 0.052 | 5 | 18.028 | 0.021 | 5 | NaN | NaN | NaN |
| UAN998005 | 2005.488 | 2005 | 6 | 28 | 18.36 | 0.098 | 18.345 | 0.175 | 14 | 18.413 | 0.095 | 18 | 18.32 | 0.025 | 6 |
| UAN998006 | 2005.759 | 2005 | 10 | 5 | 18.653 | 0.122 | 18.653 | 0.122 | 8 | NaN | NaN | NaN | NaN | NaN | NaN |
| UAN998195 | 2006.11 | 2006 | 2 | 10 | 18.873 | 0.134 | 18.905 | 0.204 | 14 | 18.853 | 0.173 | 15 | 18.862 | 0.025 | 6 |
| G-139 | 2006.756 | 2006 | 10 | 4 | 19.695 | 0.12 | NaN | NaN | NaN | 19.695 | 0.12 | 6 | NaN | NaN | NaN |
| UAN998425 | 2006.797 | 2006 | 10 | 19 | 19.65 | 0.042 | NaN | NaN | NaN | 19.696 | 0.043 | 6 | 19.603 | 0.042 | 4 |
| UAN998852 | 2006.942 | 2006 | 12 | 11 | 19.784 | 0.094 | 19.778 | 0.134 | 5 | 19.791 | 0.054 | 4 | NaN | NaN | NaN |
| UAN998898 | 2007.348 | 2007 | 5 | 8 | 20.132 | 0.098 | NaN | NaN | NaN | 20.132 | 0.098 | 5 | NaN | NaN | NaN |
| UAN999276 | 2007.89 | 2007 | 11 | 22 | 20.733 | 0.181 | NaN | NaN | NaN | 20.726 | 0.308 | 5 | 20.741 | 0.054 | 3 |
| UAN999627 | 2008.347 | 2008 | 5 | 7 | 21.198 | 0.014 | NaN | NaN | NaN | NaN | NaN | NaN | 21.198 | 0.014 | 3 |
| UAN999756 | 2008.612 | 2008 | 8 | 12 | 21.365 | 0.067 | NaN | NaN | NaN | 21.364 | 0.086 | 6 | 21.365 | 0.049 | 3 |

| | | | | | | | | | | | | | | | |
|---|---|---|---|---|---|---|---|---|---|---|---|---|---|---|---|
| UAN20100047 | 2008.956 | 2008 | 12 | 16 | 21.773 | 0.071 | NaN | NaN | NaN | 21.773 | 0.071 | 5 | NaN | NaN | NaN |
| UAN20100609 | 2009.175 | 2009 | 3 | 6 | 21.836 | 0.09 | NaN | NaN | NaN | 21.851 | 0.118 | 8 | 21.821 | 0.062 | 3 |
| UAN20101456 | 2009.567 | 2009 | 7 | 27 | 22.051 | 0.031 | NaN | NaN | NaN | NaN | NaN | NaN | 22.051 | 0.031 | 3 |

**Supplementary Material (3): Emissions Inventories**

HFC-23 (trifluoromethane, fluoroform, $CHF_3$) is a by-product of the chemical process to manufacture HCFC-22 (chlorodifluoromethane, $CHClF_2$) from chloroform and hydrogen fluoride.

**S3.1. HCFC-22 Production**

HCFC-22 is used in two ways: the commercial product is used in the refrigeration and air conditioning industries, and is eventually emitted into the atmosphere; production and consumption for this are controlled under the Montreal Protocol . It is also a chemical feedstock, the raw material for the manufacture of PTFE (polytetrafluoroethylene) and other fluoropolymers, effectively being destroyed in the process with small, inadvertent emissions not controlled under the Montreal Protocol.

Table S3 shows the inventory of HCFC-22 production for all end uses, subdivided between developed countries (referred to in the Montreal Protocol as "non-Article 5 countries", that are not eligible for any support under the Montreal Protocol or United Nations Framework Convention on Climate Change (UNFCCC) mechanisms, and individual Article 5 countries that are eligible to receive support to reduce emissions of HFC-23.

In the case of the non-A5 countries (which are listed individually in Table S4), historical demand for dispersive uses was taken from the AFEAS database [1] up to 2007 and demand for fluoropolymer feedstock was derived from Stanford Research Institute data [2] that shows historical linear growth at 5800 tonnes/year from 2001 onwards and a requirement of about 50% of the reported dispersive demand up to that date. Production for dispersive use in 2008 was derived from the Parties submissions to the Montreal Protocol [3] and the Technology and Economic Assessment Panel of the Montreal Protocol [4]. From 2009 onwards, the total production reported to the Executive Committee of the Montreal Protocol was used [5].

The same report to the Executive Committee [5] was used for production from individual Article 5 countries from 2009 onwards; prior to that year, the quantities produced in Argentina, India, South Korea, Mexico and Venezuela were estimated using the Montreal Protocol and TEAP data [3, 4].

**Table S3. Estimated HCFC-22 production: Total for all uses Gg.**

| Year | Non-Article 5 Countries | Article 5 Countries | | | | | | | Global Total |
| | | Argentina | China | India | Korea (N) | Korea (S) | Mexico | Venezuela | |
|---|---|---|---|---|---|---|---|---|---|
| 1990 | 320.57 | 0 | 0 | 3.62 | 0 | 1.75 | 1.54 | 1.85 | 329.33 |
| 1990 | 320.57 | 0 | 0 | 3.62 | 0 | 1.75 | 1.54 | 1.85 | 329.33 |
| 1991 | 355.22 | 0 | 0 | 3.86 | 0 | 2.65 | 1.84 | 1.80 | 365.37 |
| 1992 | 368.57 | 0 | 0 | 3.72 | 0 | 3.97 | 1.86 | 2.04 | 380.16 |
| 1993 | 360.93 | 0.18 | 4.92 | 4.72 | 0 | 4.41 | 2.82 | 2.01 | 379.99 |
| 1994 | 359.17 | 0.21 | 9.83 | 4.50 | 0 | 4.51 | 2.14 | 1.43 | 381.79 |
| 1995 | 365.20 | 0.00 | 14.75 | 5.22 | 0 | 5.09 | 1.96 | 1.45 | 393.67 |
| 1996 | 406.86 | 0.00 | 19.66 | 4.54 | 0 | 8.27 | 4.80 | 1.39 | 445.53 |
| 1997 | 376.66 | 0.00 | 24.58 | 5.33 | 0 | 9.28 | 4.67 | 1.37 | 421.89 |
| 1998 | 391.76 | 0.00 | 37.14 | 8.34 | 0 | 7.88 | 3.42 | 0.95 | 449.48 |
| 1999 | 378.56 | 0.00 | 59.74 | 8.68 | 0 | 14.42 | 4.89 | 1.07 | 467.37 |
| 2000 | 365.77 | 0.11 | 77.79 | 11.18 | 0 | 11.29 | 3.43 | 1.20 | 470.77 |
| 2001 | 345.20 | 0.11 | 111.42 | 12.01 | 0 | 5.81 | 2.59 | 1.32 | 478.46 |
| 2002 | 331.76 | 0.58 | 103.37 | 11.26 | 0 | 10.22 | 3.81 | 1.44 | 462.44 |
| 2003 | 326.63 | 1.06 | 144.22 | 13.88 | 0 | 6.84 | 3.70 | 1.57 | 497.89 |
| 2004 | 334.73 | 1.54 | 191.06 | 17.99 | 0 | 5.53 | 3.73 | 1.69 | 556.26 |
| 2005 | 327.37 | 2.01 | 270.89 | 17.41 | 0 | 7.92 | 5.53 | 1.81 | 632.93 |
| 2006 | 322.29 | 2.49 | 325.28 | 21.06 | 0 | 5.23 | 7.33 | 1.94 | 685.61 |
| 2007 | 328.10 | 2.96 | 414.97 | 29.32 | 0 | 4.94 | 9.13 | 2.06 | 791.47 |
| 2008 | 333.92 | 3.44 | 373.17 | 38.49 | 0 | 5.93 | 10.93 | 2.18 | 768.05 |
| 2009 | 195.80 | 3.91 | 483.98 | 47.66 | 0.50 | 6.91 | 12.73 | 2.31 | 753.80 |
| 2010 | 229.86 | 4.25 | 549.27 | 47.61 | 0.50 | 7.63 | 12.62 | 2.17 | 853.91 |
| 2011 | 241.78 | 4.02 | 596.98 | 48.48 | 0.48 | 7.26 | 11.81 | 2.44 | 913.26 |
| 2012 | 219.91 | 4.19 | 644.49 | 48.18 | 0.52 | 5.70 | 7.87 | 2.91 | 933.77 |
| 2013 | 193.52 | 1.95 | 615.90 | 40.65 | 0.58 | 6.67 | 7.38 | 2.20 | 868.86 |
| 2014 | 210.04 | 2.29 | 623.90 | 54.94 | 0.53 | 6.83 | 9.21 | 1.57 | 909.30 |
| 2015 | 225.16 | 2.45 | 534.93 | 53.31 | 0.50 | 7.18 | 4.75 | 0.68 | 828.95 |

Chinese production now accounts for 65% of the global total, with a large demand for fluoropolymer feedstock, and was estimated separately. Production for dispersive uses and export was derived from the submission to the Montreal Protocol database and TEAP data [3, 4]. Fluoropolymer (mainly polytetrafluoroethylene, PTFE) production from 1998 to 2002 was reported in China Chemical Reporter (CCR) [6] and showed growth of 33%/year. This growth was assumed to be maintained until 2007, implying production of over 69 Gg/year of PTFE in 2007, a value consistent with the capacity for fluoropolymers stated in the 11th Chinese 5 year plan to be 80 Gg/year in 2007/8 [7]. Total production of HCFC-22 in China was also reported in CCR [6], with a growth rate of between 47% and 25% in the period 1998 to 2001. For the values calculated here, a subsequent growth rate of 15% / year was applied until 2008, and from 2009 onwards, the total annual productions reported to the Executive Committee of the Montreal Protocol were used [5]. The resulting values agree within 4% with the numbers for 2013 to 2015 reported separately by the Chinese government [8].

**S3.2. HFC-23 Emissions**

Attempts to reduce HFC-23 formation by adjusting process conditions have important economic consequences for HCFC-22 production; the historic rate of HFC-23 production from a plant optimised for HCFC-22 production is 4% [9]. In plants, constructed in the last 10 years, this has been reduced to about 3% [5]. HFC-23 has few uses, some of which (for example, as a fire suppressing agent) will result in the eventual emission of most or all into the atmosphere. In the 21st century emissions from these uses have been almost constant at $133 \pm 9$ metric tonnes year$^{-1}$, a maximum of 10% of all emissions [10]. Prevention of emissions of HFC-23 requires the capture and treatment of the process vent stream, generally accomplished by high temperature oxidation.

Developed country signatories to the United Nations Framework Convention on Climate Change (UNFCCC), essentially the same set as the non-A5 countries, are required to report emissions of each HFC greenhouse gas each year. The emissions reported by individual countries are shown in the first columns of Table S4 [10]; changes in accounting procedure, such as happened in Germany from 2007 were accommodated by using the original contemporaneous data files (rather than the compendia published in 2017). This is consistent with the step changes, that resulted either from closure of the HCFC-22 production facility or from capture and thermal oxidation of the HFC-23, and with pollutant reports to national authorities [11, 12].

The second set of columns in Table S4 shows the estimated emissions of HFC-23 from those countries that are eligible for assistance under the Clean Development Mechanism (CDM) of the UNFCCC. Essentially, this rewarded destruction of HFC-23 at 11700 times the value of the same mass of $CO_2$, a gearing ratio that distorted the economics of HCFC-22 production [13] and led to the closure of the CDM to HFC-23 projects after 2009. The decision of the EU to ban the use of HFC-23 certified emission reduction (CER) credits in the European Union Emissions Trading System from 1 May 2013 effectively rendered these CERs valueless [5].

The emissions in Table S4 were calculated by estimating the annual production of HFC-23 for each country and then subtracting the quantity estimated to have been abated.

Argentina and Mexico - from 1990 to 2011, production of HFC-23 was estimated at 3.6%, falling to 3% of HCFC-22 production; from 2012 to 2015, the actual productions reported by the Executive Committee of the Montreal Protocol [5] were used. This was abated up to the maximum claimed under the CDM [14] up to May 2013, after which the destruction facilities were apparently shut down and the HFC-23 was released into the atmosphere [5].

China - from 1990 to 2006, a production rate of 3.6% was assumed, falling to 2.8% subsequently [5]. Abatement at the maximum rate allowed for the 11 of 32 plants operating under the CDM was then assumed until 2012 with the other 21 plants operating without abatement. From 2012 onwards, the actual emissions reported by China were used [5]. The quantities of HFC-23 destroyed in the period 2007 to 2015 varied between 28 and 47% of that produced.

India - up to year 2000, a production rate of 3.6% was assumed, which then dropped to 2.9%. Apparently, all of the India plants have abatement technology and, after 2006, no emissions were estimated.

South Korea - a production rate of 4%, falling to 3% was assumed for the period 1990 to 2008. Subsequently the production reported to the Executive Committee was used [5]. This was abated at the maximum allowed within the CDM until 2012, when the destruction facility was shut down. Although the HFC-23 is recovered for sale, much of that will be emitted and this is reflected in the values shown for South Korea.

North Korea - there are no data prior to 2009 and defaults of zero have been used. From 2009 onwards, the estimates here are those given in Reference 5, with total emission.

Venezuela - the production rate throughout is set at 3%, with no abatement.

**Table S4.  National HFC-23 Emissions (Metric tonnes Mt or Mg)**

| Year | Countries Reporting to UNFCCC under CRF | | | | | | | | | | | | Countries Reporting Data under CDM | | | | | | | Total Annual Emission Gg |
|---|---|---|---|---|---|---|---|---|---|---|---|---|---|---|---|---|---|---|---|---|
| | Australia | Canada | France | Germany | Greece | Italy | Japan | Netherlands | Russia | Spain | UK | USA | Argentina | China | India | Korea (N) | Korea (S) | Mexico | Venezuela | |
| 1990 | 48.1 | 65.6 | 142.0 | 373.4 | 79.9 | 30.0 | 717.6 | 378.8 | 2428.2 | 205.4 | 972.3 | 3127.6 | 0.0 | 0.0 | 130.4 | 0.0 | 70.0 | 55.5 | 55.4 | 8.88 |
| 1991 | 96.3 | 71.4 | 184.6 | 342.9 | 94.6 | 30.0 | 1140.3 | 295.0 | 2312.8 | 186.2 | 1012.3 | 2812.0 | 0.0 | 0.0 | 138.9 | 0.0 | 106.1 | 66.1 | 54.0 | 8.94 |
| 1992 | 93.2 | 56.1 | 173.7 | 342.4 | 77.6 | 30.0 | 1185.7 | 378.0 | 1904.6 | 236.1 | 1052.4 | 3123.9 | 0.0 | 0.0 | 134.0 | 0.0 | 158.6 | 67.1 | 61.2 | 9.07 |
| 1993 | 106.9 | 0.0 | 177.5 | 342.1 | 137.3 | 30.0 | 1173.0 | 424.2 | 1234.3 | 193.0 | 1092.4 | 2846.9 | 5.4 | 176.9 | 170.0 | 0.0 | 176.4 | 101.4 | 60.4 | 8.45 |
| 1994 | 96.5 | 0.0 | 79.4 | 342.6 | 183.2 | 30.0 | 1248.8 | 544.2 | 1044.1 | 295.5 | 1132.5 | 2716.1 | 6.3 | 353.9 | 162.1 | 0.0 | 180.4 | 77.0 | 42.8 | 8.54 |
| 1995 | 65.4 | 0.1 | 19.5 | 302.8 | 278.0 | 30.8 | 1432.0 | 503.0 | 1042.3 | 396.5 | 1192.6 | 2843.7 | 0.0 | 530.8 | 187.9 | 0.0 | 223.1 | 70.4 | 43.5 | 9.16 |
| 1996 | 30.7 | 0.1 | 32.9 | 263.6 | 320.2 | 1.2 | 1422.3 | 611.9 | 917.5 | 432.4 | 1220.7 | 2690.9 | 0.0 | 707.8 | 163.4 | 0.0 | 330.9 | 172.9 | 41.7 | 9.36 |
| 1997 | 0.0 | 0.9 | 31.6 | 254.7 | 338.9 | 1.6 | 1328.5 | 621.8 | 1212.3 | 495.9 | 1330.8 | 2601.3 | 0.0 | 884.7 | 192.0 | 0.0 | 278.3 | 168.0 | 41.1 | 9.78 |
| 1998 | 0.0 | 0.3 | 20.6 | 246.5 | 373.6 | 2.5 | 1245.3 | 711.3 | 1468.2 | 437.1 | 1030.2 | 3411.2 | 0.0 | 1337.0 | 300.2 | 0.0 | 166.1 | 122.9 | 28.5 | 10.90 |
| 1999 | 0.1 | 0.4 | 38.7 | 233.5 | 430.9 | 2.5 | 1233.2 | 318.5 | 1523.2 | 511.5 | 409.9 | 2636.6 | 0.0 | 2175.8 | 312.6 | 0.0 | 311.2 | 176.2 | 32.2 | 10.35 |
| 2000 | 0.1 | 0.5 | 31.9 | 109.1 | 321.7 | 3.0 | 1149.7 | 223.8 | 1783.7 | 557.2 | 219.1 | 2468.6 | 3.4 | 2827.1 | 402.6 | 0.0 | 276.7 | 123.4 | 35.9 | 10.54 |
| 2001 | 0.1 | 0.5 | 33.0 | 99.6 | 275.1 | 3.4 | 933.7 | 41.9 | 1680.8 | 270.0 | 196.8 | 1702.6 | 3.2 | 4036.4 | 353.0 | 0.0 | 47.6 | 93.1 | 39.6 | 9.81 |
| 2002 | 0.1 | 0.5 | 34.2 | 110.1 | 277.2 | 3.9 | 667.1 | 62.4 | 1268.3 | 120.4 | 165.4 | 1819.0 | 17.5 | 3780.5 | 330.9 | 0.0 | 145.0 | 137.3 | 43.3 | 8.98 |
| 2003 | 0.1 | 0.6 | 23.4 | 53.8 | 232.8 | 4.6 | 527.7 | 37.8 | 933.5 | 176.0 | 158.9 | 1066.3 | 31.8 | 5274.9 | 408.0 | 0.0 | 145.2 | 133.0 | 47.0 | 9.26 |
| 2004 | 0.2 | 0.6 | 30.3 | 53.2 | 224.1 | 5.3 | 261.6 | 31.9 | 1160.8 | 98.8 | 29.4 | 1488.3 | 46.1 | 6945.8 | 365.0 | 0.0 | 46.0 | 134.3 | 50.7 | 10.97 |
| 2005 | 0.2 | 0.6 | 35.4 | 53.8 | 191.4 | 6.0 | 85.5 | 17.6 | 1217.7 | 92.2 | 28.0 | 1368.9 | 60.3 | 9916.5 | 50.1 | 0.0 | 117.5 | 199.0 | 54.4 | 13.50 |
| 2006 | 0.2 | 0.6 | 42.3 | 35.9 | 7.7 | 6.7 | 90.3 | 25.0 | 1045.8 | 109.4 | 17.2 | 1201.1 | 74.6 | 11132.2 | 0.0 | 0.0 | 37.0 | 0.0 | 58.1 | 13.88 |
| 2007 | 0.3 | 0.6 | 26.8 | 10.6 | 11.6 | 7.5 | 63.8 | 21.8 | 943.0 | 98.8 | 8.3 | 1470.0 | 0.0 | 7872.6 | 0.0 | 0.0 | 28.1 | 33.5 | 61.8 | 10.66 |
| 2008 | 0.3 | 0.6 | 29.0 | 9.5 | 12.6 | 8.3 | 71.5 | 19.2 | 955.9 | 101.7 | 4.7 | 1180.3 | 0.0 | 5726.7 | 0.0 | 0.0 | 57.8 | 82.7 | 65.5 | 8.33 |
| 2009 | 0.3 | 0.5 | 15.2 | 8.2 | 12.9 | 8.4 | 36.7 | 13.8 | 571.9 | 90.9 | 3.8 | 473.3 | 0.0 | 7848.3 | 0.0 | 9.1 | 87.4 | 126.6 | 69.2 | 9.38 |
| 2010 | 0.3 | 0.6 | 11.6 | 7.8 | 15.0 | 9.0 | 9.6 | 34.9 | 572.8 | 122.9 | 1.1 | 559.1 | 0.0 | 9165.0 | 0.0 | 9.0 | 109.0 | 124.0 | 65.0 | 10.82 |
| 2011 | 0.4 | 0.7 | 7.5 | 8.3 | 13.3 | 9.3 | 6.1 | 15.0 | 317.1 | 78.2 | 1.0 | 607.9 | 0.0 | 9961.3 | 0.0 | 8.6 | 97.9 | 104.4 | 73.3 | 11.31 |
| 2012 | 0.4 | 0.7 | 8.0 | 7.8 | 14.9 | 9.2 | 4.3 | 11.3 | 637.1 | 69.9 | 0.9 | 386.2 | 0.0 | 10753.9 | 0.0 | 8.4 | 51.1 | 8.1 | 87.4 | 12.06 |
| 2013 | 0.4 | 0.7 | 9.1 | 7.4 | 15.1 | 9.4 | 4.3 | 17.5 | 798.7 | 60.6 | 1.0 | 290.1 | 29.3 | 10841.1 | 0.0 | 10.6 | 100.1 | 88.0 | 66.1 | 12.35 |
| 2014 | 0.5 | 0.6 | 9.2 | 7.2 | 12.2 | 9.6 | 5.2 | 3.9 | 912.2 | 55.7 | 1.2 | 364.2 | 68.6 | 12492.5 | 0.0 | 7.8 | 205.0 | 202.8 | 47.0 | 14.41 |
| 2015 | 0.5 | 0.6 | 9.3 | 6.7 | 11.9 | 9.8 | 6.7 | 9.1 | 665.6 | 46.4 | 1.4 | 313.0 | 73.4 | 7481.8 | 0.0 | 7.4 | 204.0 | 100.8 | 20.3 | 8.97 |

**Notes.**

[1] Production of HCFC-22 up to 2007 in non-Article 5 countries downloadable from *https://agage.mit.edu/data/agage-data*

[2] Stanford Research Institute, International, 1998: Fluorocarbons, Sections 543.7001 to 543.7005 of *Chemical Economics Handbook*, SRI International, Menlo Park, USA, updated using Will R. and H. Mori, Fluorocarbons, Chemical Economics Handbook 543.7000 of SRI Consulting, Access Intelligence (*www.sriconsulting.com*), 2008.

[3] Production and Consumption of Ozone Depleting Substances under the Montreal Protocol, 1986-2015, *United Nations Environment Programme*, available at *http://ozone.unep.org/en/data-reporting/data-centre*

[4] UNEP 2006 Assessment Report of the Technology and Economic Assessment Panel, *United Nations Environment Programme*, Nairobi, 2006.

[5] Key aspects related to HFC-23 by-product control technologies (Decision 78/5), Report to the Executive Committee of the Multilateral Fund for the Implementation of the Montreal Protocol, UNEP/OzL.Pro/ExCom/79/48 of 7 June 2017 available at *ozone.unep.org*

[6] Market Report: Fluorochemical develops rapidly in China, *China Chemical Reporter, 13*, Sep 6, 2002.

[7] Development and Forecast Report on China Fluorine Industry between 2007 and 2008, *www.acunion.net,* 2009.

[8] Wang Kaixiang, HCFCs/HFCs Production in China, Foreign Economic Cooperation Office, FECO/MEP, May 2015.

[9] Intergovernmental Panel on Climate Change. Revised 1996 Guidelines for National Greenhouse Gas Inventories, Reference manual, vol 3, *IPCC/IGES*, Kanagawa, Japan, 1996.

[10] Data reported under the *Common Reporting Format* and in *National Inventory Reports* available at *http://unfccc.int/national_reports/annex_i_ghg_inventories/ national_inventories_submissions/items/10116.php*.

[11] US EPA Facility Level Greenhouse Gas Emissions Data available at *https://ghgdata.epa.gov/ghgp/main.do*

[12] European Pollutant Release and Transfer Register (E-PRTR) available at *http://prtr.ec.europa.eu*

[13] Munnings C., B. Leard and A. Bento, The net emissions impact of HFC-23 offset projects from the Clean Development Mechanism, Resources for the Future, Discussion Paper 16-01, 2016.

[14] UNFCCC, Clean Development Mechanism Project Activities available at *http://cdm.unfccc.int/Projects/Index.html*

**Supplementary Material (4): Influence of OH on the inversions.**

Small differences were found in the derived emissions of HCFC-22, whereas, owing to its very long lifetime, negligible differences were found for HFC-23.

For HCFC-22, the magnitude of the difference when Rigby et al., (2017) OH was used versus an annually repeating OH concentration was much smaller than the derived uncertainty.

[Figure]

Figure S3. Potential variations in OH concentrations on the inversions

**Supplementary Material (5): European Estimates Using FLEXPART and Empa Inversion**

**Detailed Results**

The inversion results suggest that European emissions of HFC-23 in general were larger than reported to UNFCCC and exhibited considerable year-to-year variability. *A posteriori* estimates from the two inversions using different *a priori* emissions mostly agree with each other within the scope of their uncertainty limits (see Figure 6 in the main manuscript and Figure S4). Exceptions are the Italian estimates for the years 2013 and 2015, when the use of the UNFCCC a-priori resulted in much larger *a posteriori* emissions than the use of the 'UNFCCC r0.5' *a priori*. Furthermore, a large difference was also obtained for France in 2013, again the UNFCCC inversion yielding larger *a posteriori* emissions than the UNFCCCC r0.5 inversion. All regions except Spain exhibited larger *a posteriori* than *a priori* emissions for all years. These differences were most significant for Italy where average *a posteriori* emissions of 38±10 Mg/yr were estimated for the years 2009 to 2016. Although Italian *a posteriori* emissions were relatively low and closer to the *a priori* estimate in 2016 there is no clear negative trend in the emissions. Emissions from the Benelux region grew steadily until 2013 and dropped sharply afterwards, a tendency only partly reflected in the UNFCCC estimates. French *a posteriori* emissions agreed fairly well with the UNFCCC reports, with the exception of 2013 when at least one of the inversions yielded significantly higher emissions. A similar statement can be made for the United Kingdom, where only the *a posteriori* estimates for the year 2014 deviates more strongly form the UNFCCC values. The German *a posteriori* emissions were considerably larger than the *a priori* until 2012, thereafter they were closer to the reported UNFCCC values. Our *a posteriori* estimates for the Iberian Peninsula remained relatively close to the UNFCCC *a priori*. Total emissions for the six European regions listed in Table 4 (main manuscript) ranged from 108±30 Mg/yr in 2015 up to 293±43 Mg/yr in 2013 and showed a slightly negative, but insignificant trend for the period analysed here.

Compared to previous estimates by Keller et al. (2011) the estimates in this study for the years 2009 and 2010 are similar for Italy and the Benelux region, but were considerably smaller for Germany, France and the UK. The large difference for Germany may be explained by the much larger *a priori* estimate of 50 Mg/yr in Keller et al. (2011). For France and the UK similar *a priori* values were used and the differences may result from different selection of observation data. In Keller et al. (2011) the inversion was done for observations from July 2008 to July 2010, whereas here each inversion is based on one calendar year of observations.

The model performance was analysed at both Jungfraujoch and Mace Head with respect to correlations and root mean square error of simulated versus simulated time series (Figure S5). A large part of the correlation between simulation and observation is actually due to the increasing trend in HFC-23. Therefore, the correlation of the above-baseline signal can be seen as a better metric for the model performance. The latter increased considerably from a-priori to *a posteriori* for Jungfraujoch and only slightly for Mace Head. Again, there was year-to-year variability in the correlation coefficient and for Jungfraujoch a tendency to smaller correlation coefficients for later years can be seen.

[Figure]

Figure S4: Spatial distribution of HFC-23 *a posteriori* emissions (b-i) as estimated when using the UNFCCC *a priori* emissions (a). Red crosses mark the location of past and present HCFC-22 production plants.

[Figure]

Figure S5: Regional scale transport model skills as evaluated against Jungfraujoch (top) and Mace Head (bottom) observations. *A priori* performance is shown as shaded bars and *a posteriori* performance as solid bars. (left) correlation coefficient for the complete time series, (centre) correlation coefficient for the regional (above baseline) part of the time series, (right) root mean square error.

**Supplementary Material (6): Additional HFC-23 emissions**

Table S5. Annual mean global HFC-23 ($CHF_3$) emissions derived from the AGAGE 12-box model.

| Year | HFC-23 Global annual emissions (Gg yr$^{-1}$) ±1 sigma (σ) SD. | Year | HFC-23 Global annual emissions (Gg yr$^{-1}$) ±1 sigma (σ) SD. |
|---|---|---|---|
| 1930 | 0.54 ± 2.0 | 1955 | 0.11 ± 1.4 |
| 1931 | 0.52 ± 1.4 | 1956 | 0.16 ± 1.4 |
| 1932 | 0.50 ± 1.3 | 1957 | 0.20 ± 1.4 |
| 1933 | 0.47 ± 1.3 | 1958 | 0.29 ± 1.3 |
| 1934 | 0.44 ± 1.1 | 1959 | 0.39 ± 1.3 |
| 1935 | 0.41 ± 1.1 | 1960 | 0.43 ± 1.4 |
| 1936 | 0.37 ± 1.2 | 1961 | 0.50 ± 1.4 |
| 1937 | 0.34 ± 1.2 | 1962 | 0.62 ± 1.4 |
| 1938 | 0.30 ± 1.3 | 1963 | 0.76 ± 1.3 |
| 1939 | 0.27 ± 1.3 | 1964 | 0.92 ± 1.3 |
| 1940 | 0.24 ± 1.3 | 1965 | 1.10 ± 1.4 |
| 1941 | 0.20 ± 1.4 | 1966 | 1.33 ± 1.4 |
| 1942 | 0.17 ± 1.3 | 1967 | 1.60 ± 1.4 |
| 1943 | 0.15 ± 1.4 | 1968 | 1.94 ± 1.2 |
| 1944 | 0.12 ± 1.3 | 1969 | 2.15 ± 1.4 |
| 1945 | 0.09 ± 1.5 | 1970 | 2.24 ± 1.3 |
| 1946 | 0.07 ± 1.3 | 1971 | 2.38 ± 1.2 |
| 1947 | 0.05 ± 1.3 | 1972 | 2.61 ± 1.2 |
| 1948 | 0.04 ± 1.2 | 1973 | 2.95 ± 1.2 |
| 1949 | 0.03 ± 1.3 | 1974 | 2.98 ± 1.2 |
| 1950 | 0.02 ± 1.2 | 1975 | 2.99 ± 1.2 |
| 1951 | 0.01 ± 1.2 | 1976 | 2.95 ± 1.0 |
| 1952 | 0.02 ± 1.5 | 1977 | 3.17 ± 1.0 |
| 1953 | 0.04 ± 1.2 | 1978 | 3.62 ± 1.0 |
| 1954 | 0.06 ± 1.3 | 1979 | 3.92 ± 0.8 |

We thank the Referee #1 for their time and effort in evaluating this manuscript and for their suggestions for improvements. Our responses to the points made by the reviewer are addressed on the following pages.

**Replies to Referee 1**

Overview:

Review of "Recent increases in the growth rate and emissions of HFC-23 (CHF3) and the link to HCFC-22 (CHClF2) production" by Simmonds et al.
Simmonds et al. present an estimate of the HFC-23 and HCFC-22 emissions using observations from the AGAGE network. They perform two Bayesian inversions: (1) in a global 12-box model and (2) a regional inversion for Europe using FLEXPART. Overall, the manuscript reaches scientifically interesting conclusions. However, many of the details necessary to follow the conclusions are not included. Specifically, most of the details of the forward and inverse modelling are omitted. This makes it difficult to interpret the results. Additionally, some of the conclusions seem overly speculative.

I would suggest major revisions for the manuscript.

Major comments:

2.1 Structure of the manuscript and description of the modelling
The authors spend most of the methods section explaining the measurement protocols and calibration methods. Are the measurements used here fundamentally different from the previous work using the same AGAGE measurements? It seems that this paper is, at it's core, an inverse modelling paper because the novel analysis is related to the derived emissions. However, the forward and inverse modelling is not well described. The authors spend a single paragraph explaining the 12-box model. Is the box-model including a seasonal cycle or annual concentrations? If there is not a seasonal cycle, how are the authors removing the seasonal cycle from the observations? The authors mention that they use an inter-annually repeating OH but some recent work (Rigby et al., PNAS, 2017; Turner et al., PNAS, 2017) has shown variations in OH, would this be important for the modelling here?

**Reply. We have added further detail to Section 2.6 to provide further details on the inverse method. Similarly to the reviewer's concerns regarding the length of the description of the measurements, we opted to keep this section brief, because the methods have been described in detail elsewhere (Rigby et al., 2011; Rigby et al., 2013; Rigby et al; 2014). Regarding the influence of OH variations on the inversion, small differences were found in the derived emissions of HCFC-22 (which we now show in the supplement), whereas, owing to its very long lifetime, negligible differences were found for HFC-23. For HCFC-22, the magnitude of the difference when Rigby et al (2017) OH was used versus a constant OH concentration were much smaller than the derived uncertainty (Supplementary Information).**

Questions related to the 12-box model inversion: How is the inversion done? What is the state vector? Is it annual global emissions or the emissions for each of the latitudinal boxes? Presumably the distributions are Gaussian? Are there off-diagonal covariances? How are the model-data uncertainties specified (ie., what is the model error)? Some of these details could go in a supplement, but they should be described somewhere.

**Reply. We have added this information to Section 2.6.**

Similarly, the authors present a second inverse analysis within the results section (Section 3.2). The modelling is explained in a single paragraph, yet this is a very complicated inverse model they present. In the paragraph that follows, the authors state "Considerable differences between the two inversions with different a priori emission distributions occurred on the country scale" but do not explain the different inversions, priors, etc. Supplemental Section 3 provides a good explanation of the inversion framework from Brunner and Henne and I would recommend incorporating some of that text in the manuscript, it would be very useful for the reader.

**Reply. We agree with the referee that more details of the regional inversion could have been included in the main text. However, it was the intention to keep the text as concise as possible and the applied inversion tool has frequently been used in previous studies. Nevertheless, we follow the suggestion of the referee and included the details on the inversion method in a new section (3.7) in the main text, while keeping the more comprehensive discussion of the inversion results in the supplementing material.**

2.2 Differences between this work and Miller et al., (2010)?
The authors note that "Miller et al., (2010) calculated global emissions of HCFC-23 using the same AGAGE 12-box model as used here, but with a different Bayesian inverse modelling framework." (Section 3.2). However, the authors do not seem to explain the differences between the inverse modelling frameworks. This seems like a crucial detail because the HFC-23 emissions from Miller et al. are outside the errorbars presented here (Figure 3). This comment seems to go back to my previous comment on the structure of the manuscript. The authors spent a lot of time explaining the measurements but, from my reading, it doesn't seem like the measurements are what give them different emissions.

**Reply. It is important to note that the Miller et al (2010) emissions shown in Figure 3 are bottom-up estimates. We had omitted to specify this in the figure caption and have now added this in the revised version. Small differences are observed between the top-down estimates from Miller et al (2010) and those presented here. This is because: a) we include a new firn dataset in our inversion; b) Miller et al. (2010) constrained the inversion to absolute emissions from a time-varying prior, whereas we used a much simpler prior constraint that the assumed emissions growth rate would not vary by more than ±20% of the maximum bottom-up estimate, between any two years.**

2.3 Speculative statements

There are a number of statements that seem overly speculative and it's not clear that they are supported by the analysis. Here I list two rather provocative statements:

Statement (Section 3.2): "We also note that this minimum occurred during the global financial crisis of 2007-2009 and in fact HFC-23 emissions mirror global GDP growth rates for the years before and after 2009https://data.worldbank.or/indicator.NY.GDP.MKTP.CD). We can only speculate that this may have reduced the overall demand for PTFE, thereby impacting global HCFC-22 production and the co-produced HFC-23."

**Reply. This statement has been removed.**

Statement (Conclusions): "With the support of the Chinese Government, 13 new destruction facilities at 15 HCFC-22 production lines not covered by CDM were started in 2014 (UNEP, (2017a). The timing of these new initiatives is consistent with the most recent reduction (2015-2016) of global HFC-23 emissions, although we cannot confirm a direct link."

**Reply. Respectfully, we would wish to retain this statement of fact and have slightly**

**edited the text as follows-:**

**In 2014, with the support of the Chinese Government, 13 new destruction facilities at 15 HCFC-22 production lines not covered by CDM were started (UNEP, (2017a). The time frame of these new initiatives is consistent with the most recent reduction (2015-2016) of global HFC-23 emissions.**

**We thank the Referee #2 for their time and effort in evaluating this manuscript and for their suggestions for improvements. Our responses to the points made by the reviewer are addressed on the following pages.**

**Replies to Referee 2**

This manuscript reports an update of the evolution of global mole fractions and emissions of HFC-23, an important greenhouse gas. The findings are novel and interesting and the overall work is certainly of sufficient quality for publication in ACP. I would however suggest a) clarifying the reasons for publishing this gas separately from the 2017 Simmonds paper on HFCs and HCFCs and b) the following changes:

**Reply. The principal reason for a separate paper on HFC-23 is that unlike the other HFCs this is not a replacement fluorocarbon, but a compound with a more unique history as an unavoidable by-product of HFC-22 production. As such, we felt that it deserved a separate analysis, consistent with increasing attention and interest from the atmospheric science community.**

L33-36 I'm not sure what the authors want to say here. Does this mean that the HCFC22 production process has been releasing an increasingly high fraction of HFC-23 or are other sources important, too? This should be clarified in a concise way.

**Reply. Here we are just reporting HCFC-22 mole fractions. The following sentence has been added to clarify.**

**This slowing growth is consistent with demand for HCFC-22 moving from dispersive to feedstock uses, but HFC-23 emissions are a consequence of incomplete mitigation from all HCFC-22 production.**

L47-49 So which regions are likely to be responsible for the other 98.5 %?

**Reply. We have added the following sentences.**

**The majority of the increase in global HFC-23 emissions is attributed to a delay in the adoption of mitigation technologies, predominately in China and East Asia. However, a reduction in emissions is anticipated, when the Kigali 2016 amendment to the Montreal Protocol requiring HCFC and HFC production facilities to introduce destruction of HFC-23 is fully implemented.**

L53-58 This sentence is very long. Consider splitting it up to improve readability. L75 The date of that reference is inconsistent with the one in the reference list.

**Reply.  Sentence has been split into two as requested. We also thank the referee for noting the reference error which has been corrected.**

**Hydrofluorocarbons (HFCs) have been introduced as replacements for ozone-depleting chlorofluorocarbons (CFCs) and hydrochlorofluorocarbons (HCFCs), for example, HFC-134a as a direct replacement for CFC-12. Conversely, HFC-23 is primarily an unavoidable by-product of chlorodifluoromethane HCFC-22 (CHClF$_2$) production due to the over-fluorination of chloroform (CHCl$_3$).**

L75-91 This section is a rather abrupt change from the previous and could benefit from short introductory sentence or sub-heading. I would also suggest moving to after the next paragraph.

**Reply.  Introductory sentence has been added.**

**There have been a significant number of previously published papers related to HFC-23. Oram et al., (2008)………………etc…**

L88-91 I'm not sure why the extra details are relevant. Surprisingly, the reference to Simmonds et al., 2017, who reported HCFC-22 observations and emissions from the same network (and repeatedly discussed HFC-23), is missing, as are any other papers on HCFC-22.

**Reply. Extra details have been removed as requested.**

**In addition we have moved lines 66-69 to a new paragraph (after line 90) and added the following text.**

**We have previously reported on the changing trends and emissions of HCFC-22 (Simmonds et al. 2017) and references therein.**

L126-169 There are quite a lot of technical details in this section, which do not contribute to the main messages of the paper. These methods are also well established and I suggest moving large parts of the section to the supplement to improve readability.

**Reply. We have moved the bulk of the methodology in Section 2 to the supplementary material, leaving only a brief description as follows-:**

**Ambient air measurements of HFC-23 and HCFC-22 at each site are recorded using the AGAGE GC-MS-Medusa instrument which employs an adsorbent-filled (HayeSep D) microtrap cooled to ~ -175$^o$C to pre-concentrate the analytes during sample collection of 2litres of air (Miller et al., 2008; Arnold et al., 2012). Samples are analysed approximately every 2 hours and are bracketed by measurements of quaternary standards to correct for short-term drifts in instrument response. Additional details of the analytical methodology are provided in the Supplementary material.**

L203 I would be interested to know the methodology used to calculate this "liberal estimate". Is there a reference and could more information specifically on HFC-23 and HCFC-22 be given in the supplement?

**Reply. As noted above this is a very subjective question, since we cannot really know something that we cannot measure. As we explain, we are nevertheless obliged to estimate absolute accuracy for modelling purposes and we have revised the paragraph in question in order to make these points more clearly.**

**Estimates of absolute accuracy are nevertheless needed for interpretive modelling applications, so despite the subjective nature of the question it is incumbent on those responsible for the measurements to provide an assessment of accuracy. Accordingly, we liberally estimate the absolute accuracies of these measurements as -3% to +2% for HFC-23 and ±1% for HCFC-22. The larger and asymmetric uncertainty for HFC-23 is due to its lower atmospheric and standard concentration, and to the lower stated purity of the HFC-23 reagent used to prepare the primary calibration scale, respectively.**

L277 The annual growth probably relates to all of 2016, not just the end of that year?

**Reply. Yes agreed and corrected**

L293-296 The authors should make it clear that they have already published the HCFC22 data they refer to here.

**Reply. Following text has been added.**

**These results are an update of our previously reported analysis of HCFC-22 (Simmonds et al., 2017).**

L314-316 It is not clear whether this agrees with Carpenter and Reimann (2014) or not.

**Reply. The text has been revised as follows:**

**These HFC-23 emissions estimates are slightly lower in 2006 and slightly higher in 2009 than the HFC-23 estimates of Miller et al, (2010) and Carpenter and Reimann, (2014).**

L469 Please correct: un-abated.

**Reply. Corrected.**

L779-781 As published in Simmonds et al., 2017?

**Reply.The following text has been added to the note for Table 3.**

**These HCFC-22 global emissions estimates are updates for HCFC-22 emissions reported in Simmonds et al. (2017)**

L800-816 Missing from the two figures is previously published data and the respective sections of the manuscript would also benefit from a discussion of how mole fractions compare with published data. The axes on the inset of the first figure are very hard to read and the firn air is missing from the legend.

**Reply. These two figures have been modified for additional clarity and the font size on the inset in Figure 1 has been increased.  We previously discussed mole fractions in Section 3.2 and there is only a limited amount of additional reported observations of HFC-23 mixing ratios. We have added the following text to the introduction to include references to these limited observations.**

**        Kim et al., (2010) reported HFC-23 measurements (November 2007-December 2008) at Jeju Island, Korea and also estimated regional atmospheric emissions. Most recently Fang et al., (2014, 2015) have provided top down estimates of HFC-23 emissions from China and East Asia and included observed HFC-23 mixing ratios at three stations Gosan, Korea, and Hateruma and Cape Ochi-ishi, Japan.**